# The regional distribution of resident immune cells shapes distinct immunological environments along the murine epididymis

Christiane Pleuger[1,2]*, Dingding Ai[1,2], Minea L Hoppe[1,2], Laura T Winter[1,2], Daniel Bohnert[1,2], Dominik Karl[1,2], Stefan Guenther[3], Slava Epelman[4], Crystal Kantores[4], Monika Fijak[1,2], Sarina Ravens[5], Ralf Middendorff[2,6], Johannes U Mayer[7], Kate L Loveland[8,9], Mark Hedger[8,9], Sudhanshu Bhushan[1,2†], Andreas Meinhardt[1,2,8†]

[1]Institute of Anatomy and Cell Biology, Unit of Reproductive Biology, Justus-Liebig-University Giessen, Giessen, Germany; [2]Hessian Center of Reproductive Medicine, Justus-Liebig-University of Giessen, Giessen, Germany; [3]ECCPS Bioinformatics and Deep Sequencing Platform, Max Planck Institute for Heart and Lung Research, Bad Nauheim, Germany; [4]Ted Rogers Center of Heart Research, Peter Munk Cardiac Centre, Toronto General Hospital Research Institute, University Health Network, Toronto, Canada; [5]Institute of Immunology, Hannover Medical School, Hanover, Germany; [6]Institute of Anatomy and Cell Biology, Unit of Signal Transduction, Justus-Liebig-University of Giessen, Giessen, Germany; [7]Department of Dermatology and Allergology, Philipps-University of Marburg, Marburg, Germany; [8]Centre of Reproductive Health, Hudson Institute of Medical Research, Clayton, Australia; [9]Department of Molecular and Translational Sciences, School of Clinical Sciences, Monash Medical Centre, Monash University, Clayton, Australia

*For correspondence: christiane.pleuger@anatomie. med.uni-giessen.de

†These authors contributed equally to this work

**Abstract** The epididymis functions as transition zone for post-testicular sperm maturation and storage and faces contrasting immunological challenges, i.e. tolerance towards spermatozoa vs. reactivity against pathogens. Thus, normal organ function and integrity relies heavily on a tightly controlled immune balance. Previous studies described inflammation-associated tissue damage solely in the distal regions (corpus, cauda), but not in the proximal regions (initial segment, caput). To understand the observed region-specific immunity along the epididymal duct, we have used an acute bacterial epididymitis mouse model and analyzed the disease progression. Whole transcriptome analysis using RNAseq 10 days post infection showed a pro-inflammatory environment within the cauda, while the caput exhibited only minor transcriptional changes. High-dimensional flow cytometry analyses revealed drastic changes in the immune cell composition upon infection with uropathogenic *Escherichia coli*. A massive influx of neutrophils and monocytes was observed exclusively in distal regions and was associated with bacterial appearance and tissue alterations. In order to clarify the reasons for the region-specific differences in the intensity of immune responses, we investigated the heterogeneity of resident immune cell populations under physiological conditions by scRNASeq analysis of extravascular CD45+ cells. Twelve distinct immune cell subsets were identified, displaying substantial differences in distribution along the epididymis as further assessed by flow cytometry and immunofluorescence staining. Macrophages constituted the majority of resident immune cells and were further separated in distinct subgroups based on their transcriptional profile, tissue location and monocyte-dependence. Crucially, the proximal and distal regions showed striking

differences in their immunological landscapes. These findings indicate that resident immune cells are strategically positioned along the epididymal duct, potentially providing different immunological environments required for addressing the contrasting immunological challenges and thus, preserving tissue integrity and organ function.

## Editor's evaluation

This manuscript reports important findings regarding the highly variable immune environments along the epididymis. Using multiple mouse models (bacterial infection and parabiosis between WT and Ccr2 KO) in conjunction with scRNA-seq analyses, the authors provided solid evidence supporting the notion that resident immune cells are strategically positioned along the epididymal duct, potentially providing different immunological environments required for sperm maturations and elimination of pathogens ascending the urogenital tract.

## Introduction

Within the male reproductive tract, the epididymis plays an essential role in post-testicular sperm maturation and storage. Immotile spermatozoa released from the seminiferous epithelium of the testis enter the epididymis via the efferent ducts and undergo distinct consecutive biochemical maturation processes required to gain motility and fertilization capacities (*Belleannee et al., 2011*; *Skerget et al., 2015*; *Björkgren and Sipilä, 2019*; *Barrachina et al., 2022*). The sequential maturation process is orchestrated by the pseudostratified epithelium composed of several different epithelial (principal, basal, narrow/clear) and immune cell types that creates an unique luminal milieu. The barrier function of the epididymal epithelium highly depends on epithelial integrity (*Breton et al., 2019*). Intraepithelial immune cells, particularly mononuclear phagocytes (MP), are highly abundant within the epididymal epithelium and perform a key role in the preservation of epithelial integrity (*Smith et al., 2014*).

From an immunological perspective, the epididymis performs a functionally complex role by providing an immunotolerant environment for transiting immunogenic spermatozoa, while maintaining the capacity to effectively combat invading pathogens ascending from the urethra and vas deferens. Previous investigations in rodents revealed differences in the immune reactions of opposing ends of the epididymis toward ascending bacterial infection and other local and systemic inflammatory stimuli. In this regard, the proximal regions appear to be almost unresponsive, while the distal regions are prone to intense immune responses resulting in persistent tissue damage (*Michel et al., 2016*; *Silva et al., 2018*; *Klein et al., 2019*; *Wang et al., 2019*; *Klein et al., 2020*; *Wijayarathna et al., 2020*).

As the epididymis consists of a single highly convoluted duct that meanders through structurally different regions (initial segment [IS], caput, corpus, cauda), inflammation-associated tissue damage and fibrotic remodeling result in epididymal duct obstruction which has a direct impact on the maturation and passage of sperm and, thereby, fertility. The histopathological observations in rodent models replicate many of the clinical manifestations in epididymitis patients (*Pilatz et al., 2015*; *Fijak et al., 2018*). Epididymitis in humans is mostly caused by urogenital tract infections with coliform bacteria (i.e. uropathogenic *Escherichia coli* [UPEC]) or pathogens linked to sexually transmitted diseases (e.g. *Chlamydia trachomatis*, *Pilatz et al., 2015*; *Pleuger et al., 2020*) and can effectively be treated with antibiotics. However, up to 40% of epididymitis patients exhibit a persistent sub- or infertility (*Rusz et al., 2012*), most likely due to epididymal duct stenosis/obstruction and concomitant oligo- or azoospermia. The reasons underlying differences in different immune responsiveness, with strong pro-inflammatory immune response largely confined to the cauda, are not well understood.

Within the last few decades, initial steps have been made in characterizing the immunological landscape within the epididymis and understanding how the epididymis is prepared for its immunological challenges (*Nashan et al., 1989*; *Flickinger et al., 1997*; *Serre and Robaire, 1999*; *Da Silva et al., 2011*; *Shum et al., 2014*; *Pierucci-Alves et al., 2018*; *Voisin et al., 2018*; *Battistone et al., 2020*; *Mendelsohn et al., 2020*; *Wang et al., 2021*). The murine epididymis is populated by various myeloid and lymphoid cell populations that are differentially distributed along the epididymal duct. Subsets of the MP system are the most prominent group within the epididymis and form a dense network within and around the epididymal duct, especially within the IS which is the site of spermatozoa entry

(*Da Silva et al., 2011*; *Battistone et al., 2020*). Generally, the MP system comprises multiple subsets that can share similar cell surface markers, yet possess distinct functions related to tissue homeostasis and pathogen-specific immunity. Despite accumulating information about the localization and antigen presentation and antigen-processing properties (*Da Silva et al., 2011*; *Da Silva and Smith, 2015*; *Battistone et al., 2020*; *Mendelsohn et al., 2020*), the identity of MP subgroups within the epididymis as well as the full extent of their heterogeneity is still not well understood mainly due to the general similarities between macrophage and DC subpopulations. In view of the fundamentally different immunological requirements of the epididymis, maintaining both a stable and immunotolerant microenvironment for sperm maturation in the proximal regions and the ability to mount adequate immune responses toward invading bacteria at the distal end, detailed investigation of the phenotypes, localization, and function of resident immune cells is essential.

In this regard, we hypothesized that strategically positioned resident immune cells that function as both 'scavengers' and 'guardians' create distinct immunological landscapes within epididymal regions. These, in turn, are responsible for the observed differences in the intensity of the immune responses toward infectious or inflammatory stimuli as well as for tissue homeostasis and the maintenance of epithelial function that is essential for regulating the sequential steps of sperm maturation. Therefore, in this study, we aimed to both (i) analyze the differential immune responses to UPEC-elicited epididymitis and (ii) uncover the immune diversity among epididymal regions, by using an unbiased single-cell RNA sequencing (scRNASeq) analysis complemented by flow cytometry and immunofluorescence analysis to localize identified populations in situ.

## Results

### Caput and cauda epididymides react fundamentally differently during acute bacterial epididymitis

To better understand the different immune responses within the epididymal regions and to expand on our previous studies, an experimental bacterial mouse epididymitis model was used to monitor disease progression up to 10 days post infection (p.i., *Figure 1A*). Bacteria were found in all epididymal regions (IS, caput, corpus, cauda) and in the testis 1 day p.i. (*Figure 1B*), but persisted at high numbers only for up to 10 days in the cauda (*Figure 1B*). Later time points were not examined, as it is known that bacteria are cleared toward day 30 p.i. (*Klein et al., 2019*). In line with previous reports (*Klein et al., 2020*), the caput showed no gross morphological alterations (*Figure 1C*, *Figure 1—figure supplement 1A–D*), although slight histopathological changes, including mild focal epithelial damage and minor connective tissue deposition within the interstitium, were observed 5 days p.i. (*Figure 1C*) resulting in an elevated disease score that returned to normal values at 10 days p.i. (*Figure 1D*). In accordance with previous studies from our group (*Klein et al., 2020*) severe tissue remodeling was seen in the cauda (*Figure 1E*, *Figure 1—figure supplement 2*) characterized by infiltration of immune cells, loss of epithelial integrity, connective tissue deposition in the interstitium, reduction of luminal diameter, and ultimately, epididymal duct destruction resulting in a significantly increased and persistent overall disease score (*Figure 1F*). Initially, immune cell infiltrates were predominantly located peripherally within the cauda (5 days p.i.) before larger leukocytic conglomerates/granulomas developed within the entire cauda region (*Figure 1—figure supplement 2*). Sham control mice initially showed histopathological alterations in the cauda that were milder than in infected animals and returned to a level comparable to untreated epididymis toward day 10 p.i. (*Figure 1E and F*, *Figure 1—figure supplement 2*).

### Whole transcriptome and tissue analysis reveal fundamentally different immune responses in caput and cauda epididymides following infection

Initial examination pointed to different gene signatures in caput and cauda epididymides under physiological conditions, but examination under infectious conditions was not performed (*Klein et al., 2019*). We employed whole transcriptome analysis by RNA sequencing of total caput (including the IS), corpus and cauda 10 days p.i. to investigate the principal changes in the whole transcriptome of the different epididymal regions under pathological conditions in vivo. In line with the minimal histopathological alterations, almost no transcriptional differences were identified between the caput of sham- and UPEC-infected mice (in total 5 differentially expressed genes (DEG), cut-off: FDR ≤0.05,

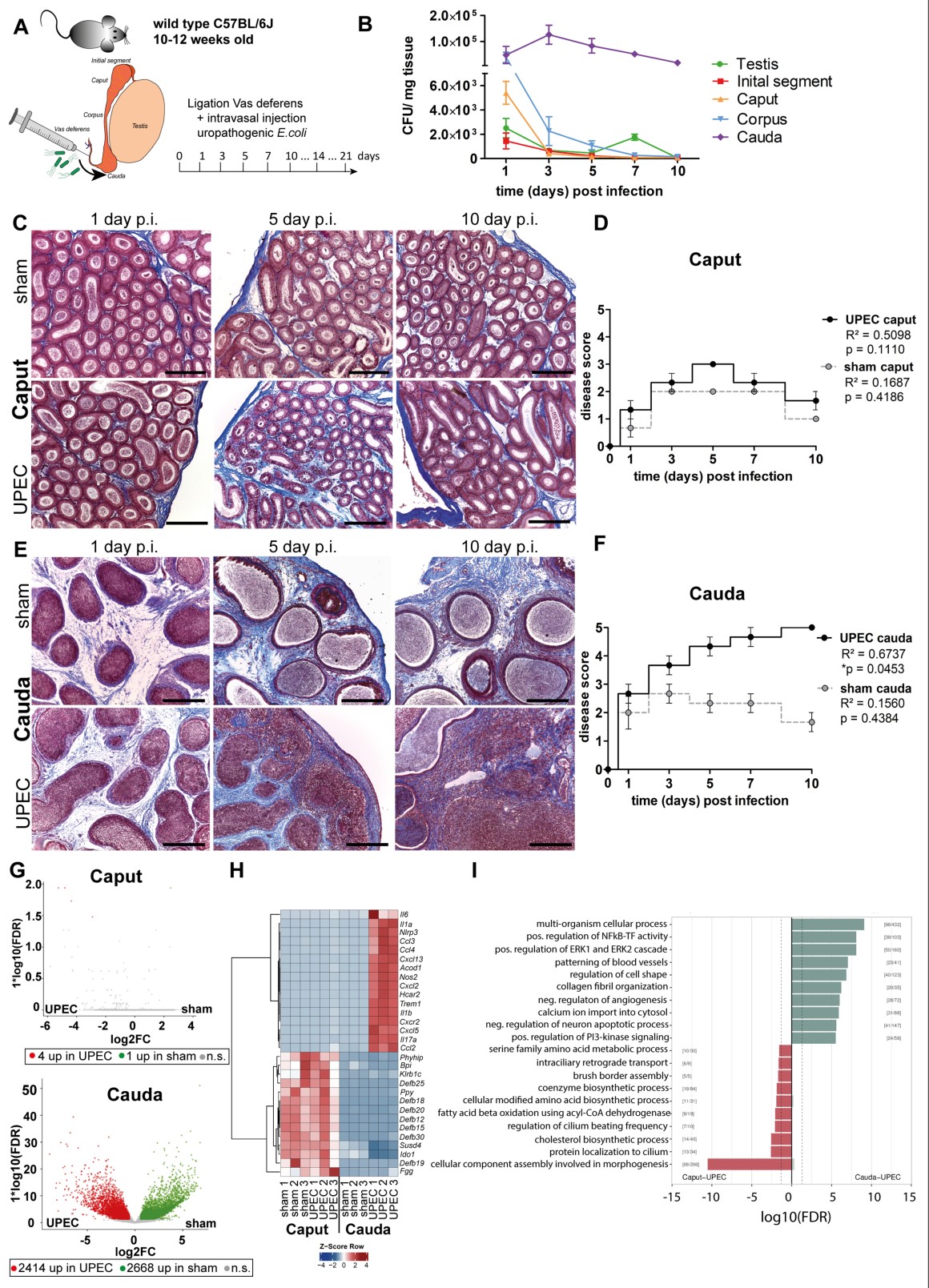

**Figure 1.** Analysis of differential immune responses of caput and cauda epididymides following uropathogenic *Escherichia coli* (UPEC) infection in C57BL/6J wild type mice. (**A**) Male C57BL/6J mice (10–12 weeks of age) were intravasally injected with UPEC or saline vehicle alone (sham) after ligation of the vas deferens. For the study organs were harvested and analyzed at the indicated time points. (**B**) Bacterial loads were assessed by determining colony forming units per mg tissue at the indicated time points within testis and the four main epididymal regions (initial segment [IS], caput, corpus,

*Figure 1 continued on next page*

*Figure 1 continued*

cauda; n=4 per time point, mean ± SD). (**C and D**) Modified Masson-Goldner trichrome staining of caput (**C**) and cauda (**D**) epididymides showing histological differences between sham- and UPEC-infected mice at day 1, day 5, and day 10 post infection. Scale bar 50 µm. (**E and F**) Pearson's correlation plot of infection time point (days post infection) and disease score of caput (**E**) and cauda (**F**). The average ± SEM disease score per time point (n=4 per time point) for sham- and UPEC-infected mice is shown. Pearson's correlation was considered to be statistically significant at p<0.05. (**G**) Volcano plot of differentially expressed genes (DEG) identified between sham- and UPEC-infected mice within caput and cauda epididymides by RNASeq analysis. Numbers of DEG are indicated below the respective plot. Cut-off criteria: FDR ≤0.05, –1 < logFC > 1. (**H**) Top 30 DEG by comparing caput and cauda epididymides of sham- and UPEC-infected mice. Cut-off criteria: FDR ≤0.05, –1 < logFC > 1. (**I**) Gene set enrichment analysis using DEG between caput and cauda epididymides of UPEC-infected mice. Cut-off criteria: FDR < 0.2, Top up/downregulated gene sets based on gene ontology.

The online version of this article includes the following figure supplement(s) for figure 1:

**Figure supplement 1.** Morphometric assessment of the differential immune responses within caput and cauda epididymides in C57BL/6J mice.

**Figure supplement 2.** Histological images (modified Masson-Goldner trichrome staining) of the epididymis of naïve, sham- and uropathogenic *Escherichia coli* (UPEC)-infected C57BL/6J mice at different time points (day 1, 5, 10 post infection).

**Figure supplement 3.** RNASeq analyses of sham- and uropathogenic *Escherichia coli* (UPEC)-infected C57BL/6J wild type mice.

–1 < logFC > 1, *Figure 1G*, *Figure 1—figure supplement 3A*). Intriguingly, although the transcriptional profiles of the caput in sham- and UPEC-infected mice were very similar, upregulation of a few infection-related genes such as *S100a8*, *S100a9*, and *Slfn4* was indicative for the presence of UPEC in the infected caput (*Figure 1—figure supplement 3B*). In contrast, the cauda of sham- and UPEC-infected mice showed considerable transcriptional differences (in total 5082 DEG, cut-off: FDS ≤0.05, –1 < logFC > 1, *Figure 1g*, *Figure 1—figure supplement 3C*). As shown by principal component analysis (PCA), the transcriptional changes in the corpus were intermediate compared with those in caput and cauda epididymides (*Figure 1—figure supplement 3A*), an observation that was reflected in a comparable magnitude of histopathological alterations (*Figure 1—figure supplement 2*). To analyze principal differences, we focused on caput and cauda in subsequent studies as these regions displayed greater differences in gene expression levels and histopathology.

Compared to the cauda, the caput was highly enriched in transcripts encoding immunomodulatory factors, such as β-defensins, bactericidal permeability-increasing protein, and indoleamine 2,3-dioxygenase 1, with no changes in the high levels observed in sham- and UPEC-infected mice (*Figure 1H*). In contrast, compared to sham control mice, the cauda of UPEC-infected mice was characterized by an upregulation of numerous transcripts encoding pro-inflammatory mediators, including pro-inflammatory cytokines (e.g. *Il-1α, Il-6, Il-17*) and chemoattractants (e.g. *Ccl2, Ccl3, Ccl4, Cxcl2, Cxcl5*) as well as inflammasome-associated transcripts (e.g. *Nlrp3, Il1b*) (*Figure 1H*, *Figure 1—figure supplement 3D*).

By grouping transcripts according to their gene ontology and pathway contribution, the cauda of UPEC-infected mice 10 days p.i. displayed upregulation of gene sets associated with fibrotic tissue remodeling and pro-inflammatory immune responses (e.g. positive regulation of NF kappa B – transcription factor activity, collagen fibril organization, and positive regulation of the ERK1 and ERK2 cascade, *Figure 1I*). Further pathway analyses revealed an upregulation of gene sets associated with B and T cell activation, indicating a transition from the innate to the adaptive immune response at this stage of infection within the cauda (*Figure 1—figure supplement 3*). The caput epididymidis of sham and infected mice were enriched with gene sets related to sperm maturation (e.g. protein localization in cilium, cellular component assembly involved in morphogenesis, and regulation of cilia beating frequency, *Figure 1I*), indicative of normal epididymal function.

## Flow cytometry analysis of immune cell populations in UPEC-infected mice

In line with the observed histopathological alterations and the transcriptional profile of the cauda of UPEC-infected mice, disease progression correlated positively with the appearance and degree of immune cell infiltration in this region (*Figure 2A*, *Figure 1—figure supplement 1D*). Notably, we observed an increase in the total immune cell population (CD45[+]) in both the caput and cauda of sham mice with most immune cell infiltrates observed at day 5, which returned to normal levels by day 14 (*Figure 2B*). This indicated that an immune response was elicited in the absence of pathogens by surgery-associated trauma and ductal pressure due to the ligation of the vas deferens. In the context

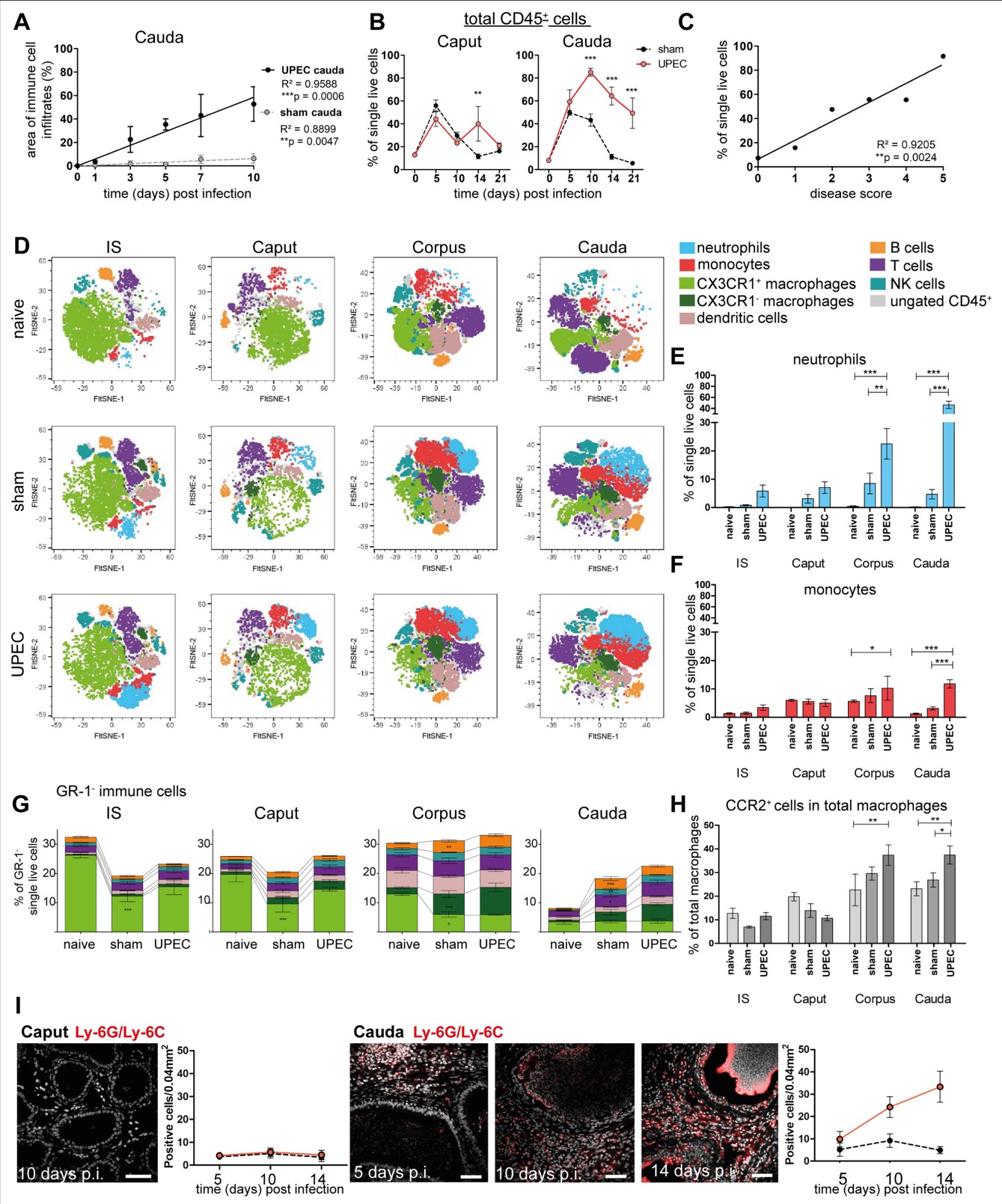

**Figure 2.** Analysis of changes in immune cell populations following infection with uropathogenic *Escherichia coli* (UPEC) in C57BL/6J wild type mice. (**A**) Pearson's correlation plot of infection time points (days post infection) and the area of immune cell infiltration within the total cauda area (%) determined by histological evaluation. Mean ± SD of at least two independent experiments with each n=4 are plotted per time point for sham- and UPEC-infected mice. Pearson's correlation was considered to be statistically significant at p<0.05 (*p<0.05, **p<0.005, ***p<0.001). (**B**) Percentage of CD45+ cells in

*Figure 2 continued on next page*

*Figure 2 continued*

single live cells within caput and cauda assessed by flow cytometry at different time points (days) post infection (mean ± SD, n=4, two-way ANOVA with Bonferroni post hoc test, *p<0.05, **p<0.005, ***p<0.001). (**C**) Pearson's correlation plot showing disease score and percentage of CD45$^+$ cells in single live cells. Pearson's correlation was considered to be statistically significant at p<0.05. (**D**) FltSNE plots of CD45$^+$ populations in naïve, sham- and UPEC-infected mice 10 days after infection. Cells were gated as described in *Figure 2—figure supplement 1* and downsampled to equal cell numbers for each segment. Samples from all biological groups (three biological replicates, respectively) were concatenated, FltSNE plots (perplexity: 20, max. iterations 1000, exaggeration factor: 12) were generated and individually gated cell populations were overlaid using FlowJo software and colored according to the legend on the right. (**E**) Bar diagram showing the ratio of neutrophils (GR-1$^+$SSC$^{hi}$ cells) within single live cells in initial segment (IS), caput, corpus, cauda of naïve, sham- and UPEC-infected mice 10 days after infection, 4–6 biological replicates from two independent experiments were grouped, mean ± SD, two-way ANOVA with Bonferroni post hoc test, *p<0.05, **p<0.005, ***p>0.001. (**F**) Bar diagram showing the ratio of monocytes (GR-1$^+$SSC$^{lo}$ cells) within single live cells in IS, caput, corpus, cauda of naïve, sham- and UPEC-infected mice 10 days after infection (4–6 biological replicates from two independent experiments were grouped, mean ± SD, two-way ANOVA with Bonferroni post hoc test, *p<0.05, **p<0.005, ***p>0.001). (**G**) Stacked bar diagrams showing the ratio of analyzed GR-1$^-$ immune cells within single live cells in IS, caput, corpus, cauda of naïve, sham- and UPEC-infected mice 10 days after infection (4–6 biological replicates from two independent experiments were grouped, mean ± SD, two-way ANOVA with Bonferroni post hoc test, *p<0.05, **p<0.005, ***p>0.001). Identified immune cells are colored equally to the FltSNE plots shown in (D). In both panels indicated immune cells were identified according to the gating strategy displayed in *Figure 2—figure supplement 1*. (**H**) Bar diagram showing the ratio of CCR2$^+$ cells in the total macrophage population (F4/80$^+$CX3CR1$^{+/-}$), 4–6 biological replicates from two independent experiments were grouped, mean ± SD, two-way ANOVA with Bonferroni post hoc test, *p<0.05, **p<0.005, ***p>0.001. (**I**) Confocal microscopy images showing the location of Ly6G$^+$Ly6C$^+$ cells (GR-1$^+$, red) within caput and cauda of UPEC-infected mice 5, 10, and 14 days post infection (nuclei in gray) including bar diagrams showing the semi-quantified summary of all immunostained tissues (by counting Ly6G$^+$Ly6C$^+$ cells within caput and cauda of sham- and UPEC-infected mice, n=4, for each biological replicate three representative areas were counted, mean ± SD). Scale bar 50 μm.

The online version of this article includes the following figure supplement(s) for figure 2:

**Figure supplement 1.** Gating strategy behind flow cytometry analyses of all immune cell populations under pathological conditions (displayed *Figure 2*).

**Figure supplement 2.** Infiltration of neutrophils in relation the bacterial appearance.

**Figure supplement 3.** Multiplex assay-based determination of cytokine levels from *ex vivo* organ culture.

of infection with UPEC, an increased infiltration of immune cells was observed in the cauda, whereas the caput showed a significant increase of immune cells compared to sham injected mice only at day 14 p.i. (*Figure 2B*). Immune cell infiltration peaked at 10 days p.i., which correlated with disease score of the infected animals (*Figure 2C*).

As the RNASeq data indicated a transition from innate to adaptive immune responses 10 days after infection, which also correlated with the peak of immune cell infiltration in several segments, we aimed to further characterize immune cell populations within all epididymal regions (IS, caput, corpus, cauda) of naive, sham- and UPEC-infected mice. We designed a flow cytometry panel that allowed us to simultaneously identify different populations of innate (neutrophils, monocytes, macrophages, dendritic cells, NK cells) and adaptive immune cells (B and T cells, *Figure 2—figure supplement 1*). CX3CR1$^+$ macrophages represented the most dominant immune cell population in the IS and caput, whereas immune cell composition was more diverse in the corpus and cauda (*Figure 2D*). While neutrophils were absent in samples from naive mice, infiltrates of Gr-1$^+$SSC$^{hi}$ neutrophils (*Figure 2E*) and GR-1$^+$SSC$^{lo}$ monocytes (*Figure 2F*) were most pronounced in the corpus and cauda upon UPEC infection (*Figure 2D–F. Figure 2—figure supplement 2*).

Furthermore, both corpus and cauda showed a significant increase in adaptive immune cell populations (B and T lymphocytes), which were present after sham injection and UPEC infection (*Figure 2G*) and correlated with the observed enrichment of gene sets associated with B and T cell activation at 10 days p.i. (*Figure 1—figure supplement 3E*).

Compared to naive mice the ratio of CX3CR1$^+$ macrophages significantly decreased in sham- and UPEC-infected mice, particularly within the proximal regions (IS and caput, *Figure 2G*). In contrast to the proximal regions, the decrease of CX3CR1$^+$ macrophages was accompanied by an increased ratio of CX3CR1$^-$ macrophages within the distal regions (corpus, cauda), indicating a shift in the macrophage pool (*Figure 2G*). Notably, we observed a significantly increased ratio of CCR2$^+$ cells within the total macrophages (CX3CR1$^+$ and CX3CR1$^-$) in the corpus and cauda of UPEC-infected mice (*Figure 2H*), which indicates a potential contribution of monocytes to the macrophage pool within distal but not proximal regions upon UPEC infection.

Overall, the corpus and cauda developed a highly inflammatory immune environment in which subgroups of innate antigen-presenting myeloid (macrophages and cDC) and effector lymphoid cells co-existed. In line with histological observations, also sham-infected mice developed an inflammatory response with similar, yet milder changes in the immune cell composition (*Figure 2D–H*).

As seen by immunofluorescence analysis, numbers of Ly6G⁺ and Ly6C⁺ cells (including neutrophils and monocytes) were progressively increasing within the interstitium of the cauda, but not the caput, and also could be identified in the epididymal epithelium (*Figure 2I*) at time points when epithelial integrity was disturbed.

## Simultaneous exposure to an inflammatory stimulus in vitro results in differential immune responsiveness of the epididymal regions

To examine whether the observed differential immune responses within epididymal regions were merely a consequence of microbial ascension and thus the longer exposure of the cauda to the pathogens, we have utilized an *ex vivo* organ culture model that allows simultaneous challenge with an inflammatory stimulus. Cytokine production profiles of the different epididymal regions (IS, caput, corpus, and cauda) were analyzed separately after stimulation with ultrapure lipopolysaccharide (LPS). While the IS and caput were still mostly unreactive, both corpus and cauda showed a significant upregulation of IL-1α, IL-1β, TNFα, MCP-1 (CCL2), IL-6, and IL-10 (*Figure 2—figure supplement 3A*). Intriguingly, IS and caput showed a higher intracellular bacteria load compared to corpus and cauda after *ex vivo* co-culture of organ pieces with UPEC, indicative for a higher and faster bacterial uptake and clearance potential (*Figure 2—figure supplement 3B*). Overall, these data suggest that the fundamentally different immunological responses observed in vivo within different regions of the epididymis are an inherent feature of the region, and thus independent of the administration route of the inflammatory stimulus.

## Single-cell transcriptomic analysis of immune cells in the epididymis demonstrates regional heterogeneity in steady state

The above described observations indicated the possibility of differential immunological landscapes in the epididymal regions. To gain a comprehensive understanding, we employed scRNASeq of extravascular CD45⁺ cells. For this purpose, C57BL/6J wild type mice were intravenously injected with an APC/Cyanine7-conjugated anti-CD45.2 antibody (*Figure 3A*) prior to killing and organ collection. This allowed a later discrimination of tissue-resident immune cells that were labeled with a PerCP-Cyanine 5.5-conjugated CD45.1 antibody from intravascular CD45.2⁺ cells (*Figure 3—figure supplement 1*). In total 12,966 cells were separately isolated from the four main epididymal regions (IS, caput, corpus, cauda). The data were subsequently combined into a single dataset to investigate their regional distribution (*Figure 3A*, *Figure 3—figure supplement 2*). Unsupervised clustering and uniform manifold approximation and projection (UMAP) identified 13 different clusters (*Figure 3B*) with distinct gene expression profiles (*Figure 3C*). The identity of each cluster was annotated manually based on key marker gene expression (*Figure 3D and E*, *Figure 3—figure supplement 3*). Among the clusters, the majority of identified immune cells comprised several myeloid cell populations comprised several myeloid cell populations and included subsets of macrophages (clusters 1, 2, and 7), monocytes (clusters 8 and 10), and dendritic cells (clusters 2, 5, and 13). Macrophages were broadly identified by the co-expression of multiple key marker genes such as *C1qa*, *Fcgr1* (encoding CD64), *Adgre1* (encoding F4/80), and *Cd68*, with alternating levels of markers such as *Cx3cr1*, *Ccr2*, *H2-Aa* (encoding an MHC-II component). Monocytes were broadly characterized by the expression of *Ly6c2*, *Ccr2*, and *Ace*. Dendritic cells (DC, *Flt3*⁺ high expression levels of MHC-II transcripts) were segregated into three clusters that were identified as conventional DC 1 (*Clec9a⁺Irf8⁺*), conventional DC 2 (*Cd209a⁺*), as well as a small population of migratory DC (*Ccr7⁺*, *Figure 3B–E*). Apart from myeloid cells, all epididymal regions were populated by lymphocytes, including T cells (*Cd3e⁺*, clusters 4 and 9), NK cells (*Nkg7⁺Eomes⁺*, cluster 6), and B cells (*Cd79a⁺*, cluster 11, *Figure 3D and E*). T cells were further discriminated into αβ and γδ T cells based on their alternating expression of *Trbc* and *Trdc*, respectively.

We next defined the cluster distribution across epididymal regions (*Figure 3F*, *Figure 3—figure supplement 3*). Transcriptomic data of identified immune cell populations and their ratios within the CD45⁺ population in different epididymal regions were subsequently confirmed at the protein level

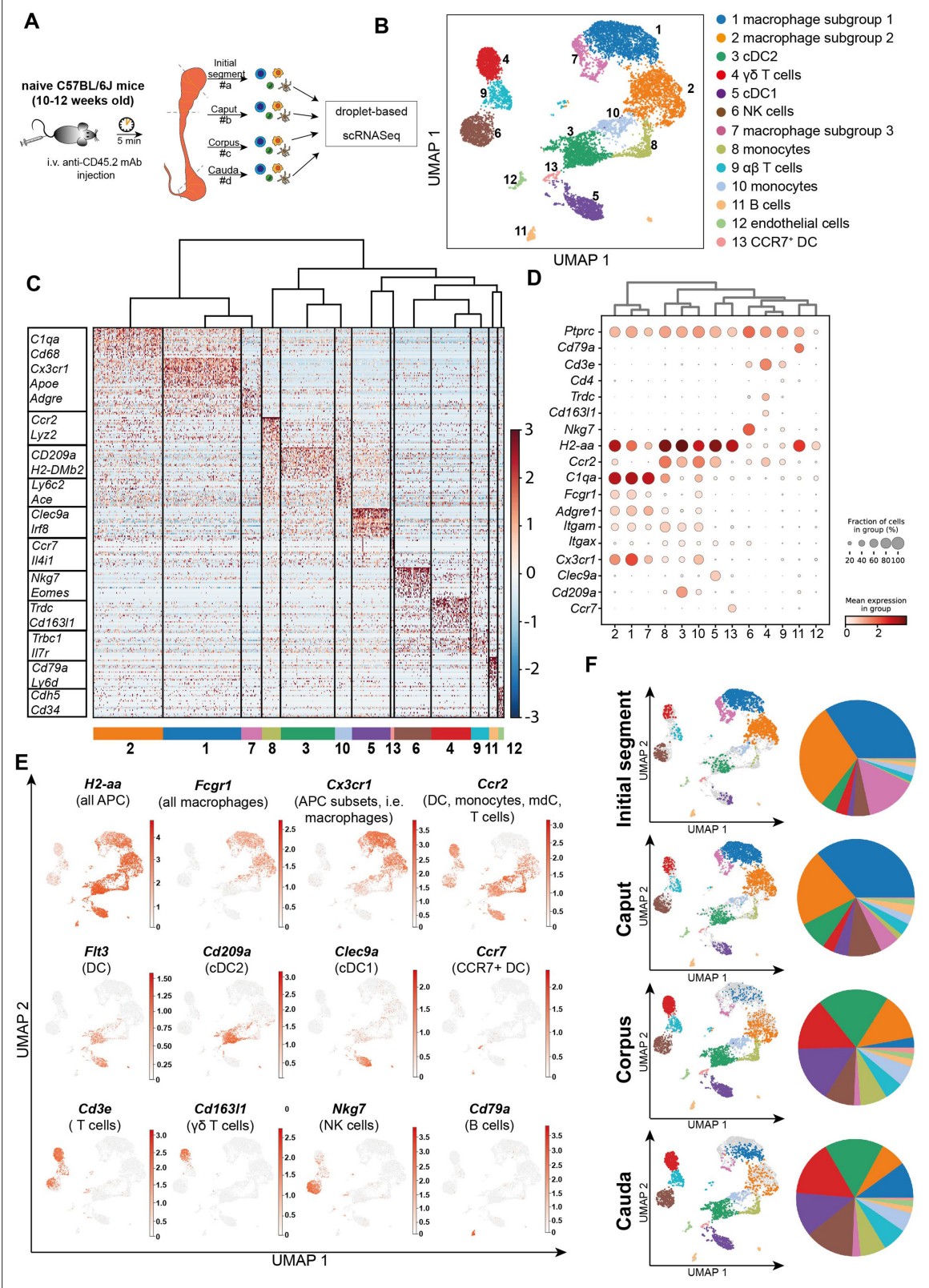

**Figure 3.** Single-cell RNA sequencing (scRNASeq) of different epididymal regions reveals immune cell heterogeneity within the murine epididymis under physiological conditions. (**A**) Schematic overview of the experimental procedure for isolating extravascular CD45+ cells from different epididymal regions. (**B**) Uniform manifold approximation and projection (UMAP) plot of 12,966 FACS-sorted CD45+ cells isolated from the four epididymal regions, showing immune cell populations identified by unsupervised clustering. (**C**) Heatmap of the Top45 marker by stringent selection of markers (only present

*Figure 3 continued on next page*

*Figure 3 continued*

in one cluster, 585 in total) showing expression differences among clusters. (**D**) Dot plot corresponding to the UMAP plot showing the expression of selected subset-specific genes – dot size resembles the percentage of cells within the cluster expressing the respective gene and dot color reflects the average expression within the cluster. (**E**) UMAP plots showing the expression of selected key markers for the indicated immune cell population (APC – antigen-presenting cells, mdC – monocyte-derived cells, DC – dendritic cells). (**F**) UMAP plots and pie charts showing regional distribution of identified clusters.

The online version of this article includes the following figure supplement(s) for figure 3:

**Figure supplement 1.** Extravascular CD45[+] cells of different epididymal regions were sorted following the indicated gating strategy prior to single-cell RNASeq.

**Figure supplement 2.** Quality controls for single-cell reads and re-confirmation of identified CD45[+].

**Figure supplement 3.** Expression of key marker genes for the identified immune cell populations within epididymal regions.

by flow cytometry (*Figure 4*, *Figure 4—figure supplement 1*, gating). The vast majority of resident immune cells in the epididymis were found in the IS (approximately 10–15% CD45[+] cells among the single live cells vs. 1–5% CD45[+] cells in single live cells in caput to cauda; *Figure 4A*). Overall, we noted similarities in the composition of resident immune cell populations in the IS and caput that were clearly distinct from that in the more distal corpus and cauda. In this regard, IS and caput were predominantly populated by macrophage subsets (approximately 78% and 66% in CD45[+] cells, respectively; *Figure 4B*) with other leukocytes accounting for <5% for each population (*Figure 4B–H*). In contrast, the corpus and cauda contained a more heterogeneous immune cell network, including several myeloid cell populations. In addition to macrophages (25–35%, *Figure 4B*), monocytes (7–10%, *Figure 4C*) and dendritic cells (cDC1 7–10% and cDC2: 12–20%, *Figure 4D and E*) were predominantly found in close conjunction with the epididymal duct, as detected by immunofluorescence analysis. In accordance with the previous studies (*Voisin et al., 2018*), no plasmacytoid dendritic cells were found within the murine epididymis.

Lymphocyte subsets (NK cells [10%, *Figure 4F*], B cells [2–5%, *Figure 4G*], T cells [10–20%, *Figure 4H*]) were located in both the interstitial and intraepithelial compartment (*Figure 4F–H*). Among the T cells, we further distinguished αβ and γδ T cells, with only the latter found within the epithelium (*Figure 4H*). The difference in leukocyte populations, their ratio, and tissue localization throughout the epididymis points to the existence of inherently different immunological environments in the proximal (IS, caput) and distal regions (corpus, cauda), which form the basis of the differential immune responsiveness observed in models of epididymitis.

## Macrophages separate into several subgroups based on their transcriptional profile

*Adgre1[+]C1qa [+]* cells, broadly considered as macrophages (*Dick et al., 2022*), constitute the majority of CD45[+] cells in the epididymis. In subsequent closer analyses with the aim to decipher the possible heterogeneity of this population, we first distinguished macrophage populations (clusters 1, 2, and 7) from monocyte populations (clusters 8 and 10) based on their expression of *C1qa*, *Ccr2*, *Ly6c2*, *Napsa*, and *Plac8* (*Figure 5A*), with both monocyte clusters expressing lower levels of *C1qa*. However, cluster 10 showed higher expression of transcripts encoding classical monocyte markers (*Ly6c2*, *Napsa*, *Plac8*) compared to cluster 8. This indicates that cluster 10 resembles a classical monocyte population, whereas cluster 8 represents a monocyte population undergoing differentiation into a macrophage phenotype. This assumption was also supported by intermediate expression of *C1qa*, *Adgre1*, and *Fcgr1* between classical monocytes (cluster 10) and macrophage populations (clusters 1, 2, and 7, *Figure 3D*, *Figure 5A*).

To further characterize the heterogeneity among macrophage subpopulations (clusters 1, 2, and 7), all cells in clusters 1, 2, and 7 were re-analyzed after exclusion of other CD45[+] cells. By unsupervised clustering, nine subgroups (*Figure 5B*) were identified, each with a distinct gene expression profile (*Figure 5C*, *Figure 5—figure supplement 1*). All identified macrophage subgroups were highly enriched with *C1qa* and *Adgre1* transcripts confirming their macrophage identity (*Figure 5D*, *Figure 5—figure supplement 1*). Clusters 1 and 2 constitute the majority of macrophages and demonstrated comparatively high expression levels of genes that were previously reported to be associated with homeostatic and sensing functions of macrophages within other tissues (i.e. brain

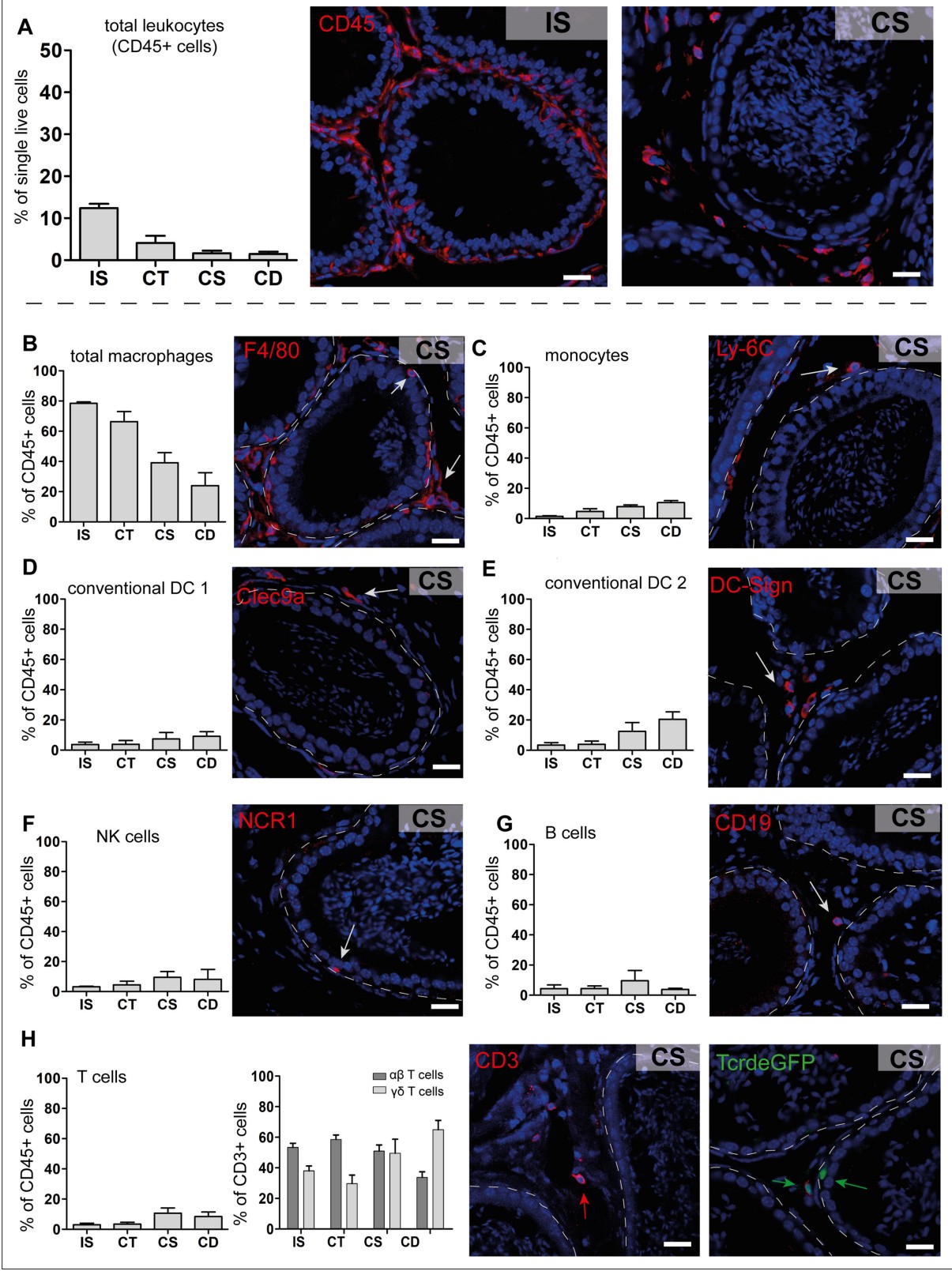

**Figure 4.** Quantification and localization of identified immune cell populations among epididymal regions (scale bar 20 µm). (**A**) Distribution (assessed by flow cytometry n=4-8, bar diagram showing mean ± SD) of total leukocytes (CD45+ cells) and localization within the initial segment and corpus, as shown by immunostaining of CD45. (**B–H**) Quantification and localization of the following immune cell populations were assessed by flow cytometry and immunostaining using selected markers (n=4–8, mean ± SD). The following markers were used: CD45, F4/80, CD11B, Ly6C, MHC-II, CLEC9A, CD209A,

*Figure 4 continued on next page*

*Figure 4 continued*

CD163, CCR2, CX3CR1 for identifying myeloid cell populations, and CD45, B220/CD45R, CD3, TCRβ, TCRγδ, NK1.1 for lymphoid cell populations (further panel information and gating strategies are displayed in the Methods section and supplemental material, respectively). Representative immunofluorescence images are displayed from the corpus (CS) regions: (**B**) total macrophages (F4/80[+], red), located in the interstitial, intraepithelial, and peritubular compartments, (**C**) monocytes (Ly-6C[+]), located in the peritubular compartment, (**D and E**) conventional dendritic cells cDC 1 (Clec9a[+]) and 2 (DC-Sign/CD209a[+]), (**F**) NK cells (NK1.1[+] for flow cytometry and NCR1 for immunostaining), located in the intraepithelial compartment, (**G**) B cells (B220/CD45R[+] for flow cytometry and CD19[+] for immunostaining), (**H**) T cells that were further segregated into αβ T cells (TCRβ[+], red) and γδ T cells (TCRγδ[+], green), scale bar 20 μm.

The online version of this article includes the following figure supplement(s) for figure 4:

**Figure supplement 1.** Gating strategy behind flow cytometry analyses of all immune cell populations under physiological conditions.

microglia, *Hickman et al., 2013*; *Van Hove et al., 2019*; *Abels et al., 2021*), indicating similar functions for the epididymis. These genes include *Cx3cr1, Tmem119, P2ry12, P2ry13, Gpr34, Rnase4, Olfml3,* and *Tgfbr1* (*Figure 5D*, *Figure 5E*). In contrast to cluster 1, cluster 2 expressed high levels of MHC-II component transcripts (e.g. *H2-Ab1*, *Figure 5D*), indicating an activated status for antigen presentation. Cluster 6 showed a similar expression pattern to cluster 2, but was highly enriched with transcripts encoding several cytokines and chemokines (*Tnf, Cxcl2, Ccl4*), as well as immediate-early response genes, such as *Fos, Jun,* and *Egr1* (*Figure 5C*, *Figure 5—figure supplement 2*). However, it is likely that the latter genes could be a consequence of tissue processing prior to sequencing (*Denisenko et al., 2020*). These transcriptional differences indicate that cluster 6 constitutes an activated form of cluster 2, hence, both clusters were considered as one subgroup.

Clusters 3 and 4 were enriched with *Ccr2* and transcripts encoding MHC-II components (e.g. *H2-Ab1, H2-Aa, H2-Eb1*, *Figure 5D*), indicating an activated pro-inflammatory phenotype. Albeit transcriptionally similar to cluster 4, cluster 3 expressed high levels of several activation genes, such as immediate-early response genes (*Fos, Jun, Egr1*, *Figure 5C*, *Figure 5—figure supplement 2*) and cytokine and chemokines (*Figure 5C*), suggesting that cluster 3 also constituted an activated subset, as was the case for cluster 6, consequently, clusters 3 and 4 were also considered as one subgroup. Clusters 5 and 9 were both enriched with *Cd163, Lyve1,* and *Folr2*, but showed reciprocal expression of *Timd4* and *Ccr2*, respectively (*Figure 5D*, *Figure 5—figure supplement 1*), indicating similar anti-inflammatory or regulatory phenotypes, but different ontogenies. Clusters 7 and 8 constituted rather minor subgroups and did not express either *Ccr2* or *Cd163, Lyve1* or *Timd4*, and only low levels of *Cx3cr1* compared to the other subgroups. In contrast to cluster 8, cluster 7 cells expressed *Trem2* beside MHC-II encoding transcripts such as *H2-Ab1* (*Figure 5D*). Cluster 8 showed a relatively high expression level of *Adgre1* compared to all other clusters in addition to high levels of *Acp5* (*Figure 5D*). The differential distribution among regions was the most striking difference among the identified subpopulations (*Figure 5F*).

## Macrophage subgroups show striking regional differences in their regional and compartmental distribution

Based on the transcriptional profiles, seven macrophage subpopulations (subgroups 1, [2+6], [3+4], 5, 7, 8, 9) were distinguished in the murine epididymis. In the next step, identified subpopulations were quantified in support by flow cytometry in wild type mice (*Figure 6—figure supplement 1* for gating strategy. *Figure 6—figure supplement 2* for ratio of subgroups in total CD45[+] cells) and localized in the tissue using immunofluorescence in *Cx3cr1*[GFP]*Ccr2*[RFP] reporter mice. Overall, F4/80[+] cells constituted approximately 80% of CD45[+] cells within the IS and these cells gradually decreased toward the cauda to approximately 25% of CD45[+] cells (*Figure 6A*). F4/80[+] cells were found throughout all epididymal regions to be constituents in both epididymal compartments, that is, the ductal epithelium and the interstitium. The majority of F4/80[+] cells were also CX3CR1 positive (*Figure 6B and C*, *Figure 6—figure supplement 3*). Only a small fraction of intraepithelial CX3CR1[+] cells was F4/80[-] within the IS (indicated by arrowheads *Figure 6C*, *Figure 6—figure supplement 3*).

In accordance with the scRNASeq data, CX3CR1[+]CCR2[-]MHC-II[-] macrophages (cluster 1) and CX3CR1[+]CCR2[-]MHC-II[+] (clusters 2 and 6) macrophages constituted the majority of F4/80[+] cells and both were highly abundant within the IS (40% of total F4/80[+] cells for each population, *Figure 6B*). Both subgroups declined toward the cauda (*Figure 6—figure supplement 2*), although the ratio of CX3CR1[+]CCR2[-]MHC-II[+] cells (cluster 2) within the F4/80[+] population remained similar throughout all

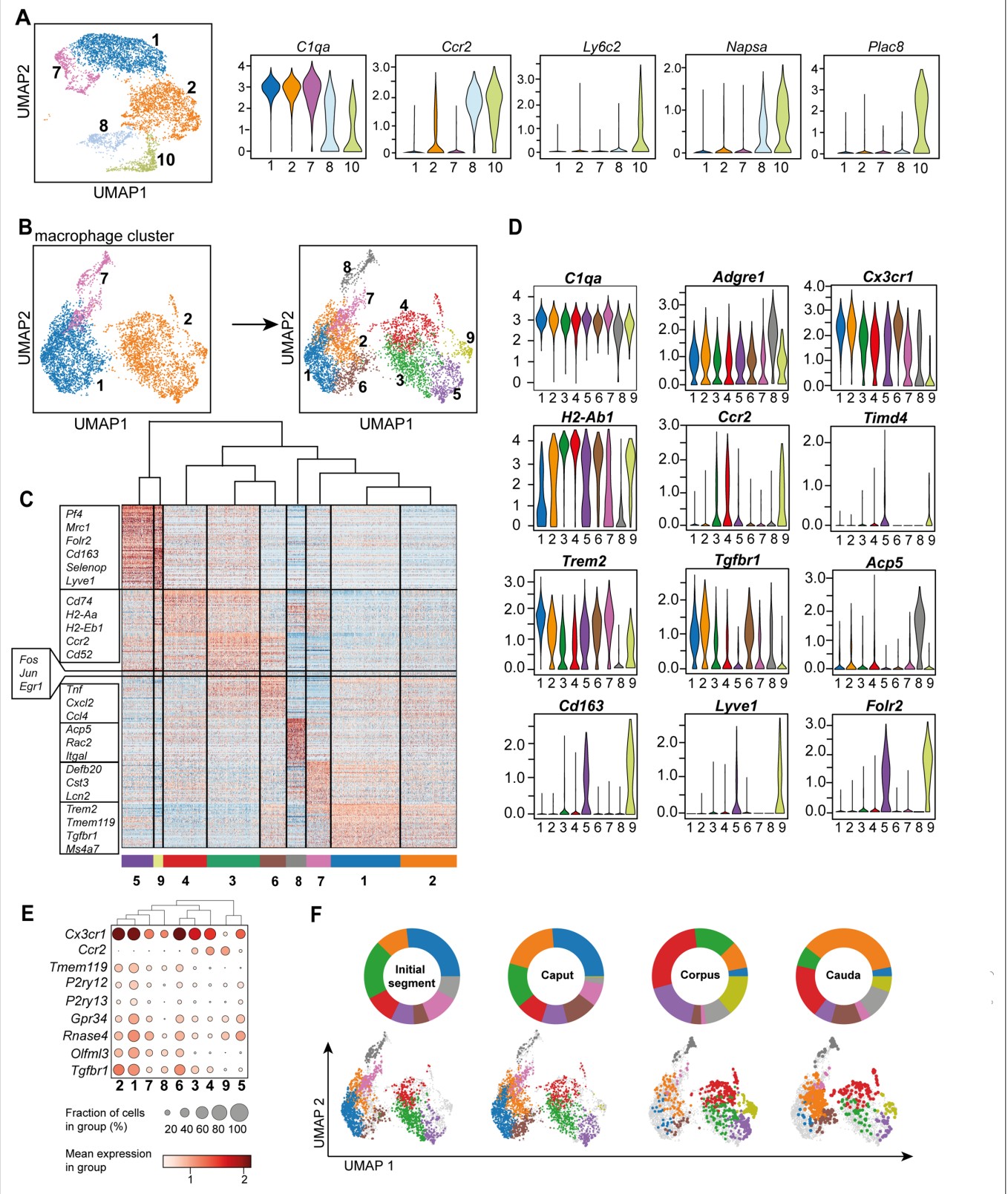

**Figure 5.** Distinct macrophage subgroups exist within the murine epididymis. (**A**) Uniform manifold approximation and projection (UMAP) plot and violin plots showing segregation of macrophages (clusters 1, 2, 7) and monocytes (clusters 8, 10) based on clustering and expression of the selected key genes *C1qa, Ccr2, Ly6c2, Napsa, Plac8*. (**B**) UMAP plot showing re-clustering of macrophage population (clusters 1, 2, 7) under exclusion of all other previously identified CD45[+] cluster resulting in the formation of nine *Adgre1*[+] subclusters. (**C**) Heatmap of the 50 most differentially expressed

*Figure 5 continued on next page*

*Figure 5 continued*

marker genes in each cluster from *Figure 4B*. (**D**) Violin plots showing the expression level of selected genes. (**E**) Dot plot corresponding to the UMAP plot showing the expression of selected subset-specific genes – dot size resembles the percentage of cells within the cluster expressing the respective gene and dot color reflects the average expression within the cluster. (**F**) UMAP plots and pie charts showing the distribution of identified macrophage populations among epididymal regions.

The online version of this article includes the following figure supplement(s) for figure 5:

**Figure supplement 1.** Uniform manifold approximation and projection (UMAP) plots showing the expression of selected key markers for identified macrophage subgroups, related to violin plots in *Figure 4D*.

**Figure supplement 2.** Violin plots showing the expression of immediate-early activation genes (*Fos, Jun, Egr1*) as well as upregulated cytokines *Tnf, Cxcl2, Ccl4* among identified macrophage subgroups.

epididymal regions (*Figure 6B*). The majority of CX3CR1$^+$CCR2$^-$MHC-II$^-$ macrophages (cluster 1) were located within the epididymal epithelium (*Figure 6C*, *Figure 6D*), between adjacent epithelial cells. Notably, while intraepithelial macrophages within the IS, which exhibited long and thin protrusions toward the lumen, did not express MHC-II, these intraepithelial cells gained MHC-II expression in caput, corpus, and cauda (indicated by arrowheads in *Figure 6D*, *Figure 6—figure supplement 4*). In addition, CX3CR1$^+$CCR2$^-$MHC-II$^+$ cells (clusters 2 and 6) closely surround the epididymal duct with highest density in the IS (*Figure 6D*, *Figure 6—figure supplement 4*).

In contrast, CX3CR1$^+$CCR2$^+$MHC-II$^+$ macrophages (clusters 3 and 4) showed the opposite distribution pattern. While being less abundant in the IS and caput (5–10%), CX3CR1$^+$CCR2$^+$MHC-II$^+$ macrophages constituted 20–30% of macrophages in the corpus and cauda (*Figure 6B*), a similar proportion to the CX3CR1$^+$CCR2$^-$MHC-II$^+$ macrophages (cluster 2). Notably, triple positive CX3CR1$^+$CCR2$^+$MHC-II$^+$ macrophages (clusters 3 and 4) were localized exclusively in the interstitium, most prominently in the cauda (*Figure 6D*, *Figure 6—figure supplement 4*).

Apart from these three major populations, the minor populations were further subdivided based on the expression of CD163 and CCR2 (*Figure 6—figure supplement 1*). Here, CD163$^+$CCR2$^-$ macrophages (cluster 5) constituted 3–10% of resident macrophages and were predominantly found in the corpus (*Figure 6B*, *Figure 6—figure supplement 2*). Similarly, CD163$^+$CCR2$^+$ macrophages (cluster 9) were most abundant in the corpus and cauda (10% of total F4/80$^+$ cells, *Figure 6B*, *Figure 6—figure supplement 2*). In tissue sections, both populations were located interstitially, whereby CD163 single positive cells (CCR2$^-$) were clustered in close proximity to vascular structures and appeared smaller in size compared to the solitarily distributed CD163$^+$CCR2$^+$ cells (*Figure 6E*). Cells that were F4/80$^+$ but CX3CR1$^-$ concomitant with the absence of CCR2 and CD163 were considered to be macrophages of clusters 7 and 8. Furthermore, cells belonging to cluster 7, but not cluster 8, express MHC-II. Both populations were most abundant within the corpus (*Figure 6B*), but constituted only a small fraction of resident immune cells (approximately 1–4% of total CD45$^+$ cells, *Figure 6—figure supplement 2*). Generally, F4/80$^+$ cells negative for both CX3CR1 and CCR2 (belonging to both clusters 7 and 8) were exclusively found within the interstitium (indicated by an asterisk in *Figure 6D*, *Figure 6—figure supplement 4*).

Overall, the regional and compartmental distribution suggests that distinct macrophage subsets populate the epididymis to facilitate the complex spectrum of canonical macrophage functions (homeostatic, inflammatory, reparative/regulatory) adapted to the needs of the respective microenvironment: 'Scavenger functions' within the epithelial compartment of the proximal regions (i.e. IS) to maintain tissue integrity, and 'guardian functions' within the distal regions to efficiently tackle invading pathogens and tissue regeneration.

## Maintenance of resident macrophages in epididymal regions depends differentially on monocyte recruitment

Having identified the variation and heterogeneity of resident macrophages among epididymal regions, we further investigated putative differences in the monocyte dependence on the maintenance of resident macrophages among the epididymal regions. Parabiosis experiments were performed by surgically conjoining CD45.1$^+$ wild type mice with CD45.2$^+$ *Ccr2$^{-/-}$* mice for 6 months before analyzing the ratio of monocyte-derived CD45.1$^+$ cells within Ly6C$^{hi}$ blood monocytes (*Figure 7A*, *Figure 7—figure supplement 1*) and resident macrophage populations of the *Ccr2$^{-/-}$* recipient mouse (*Figure 7A*).

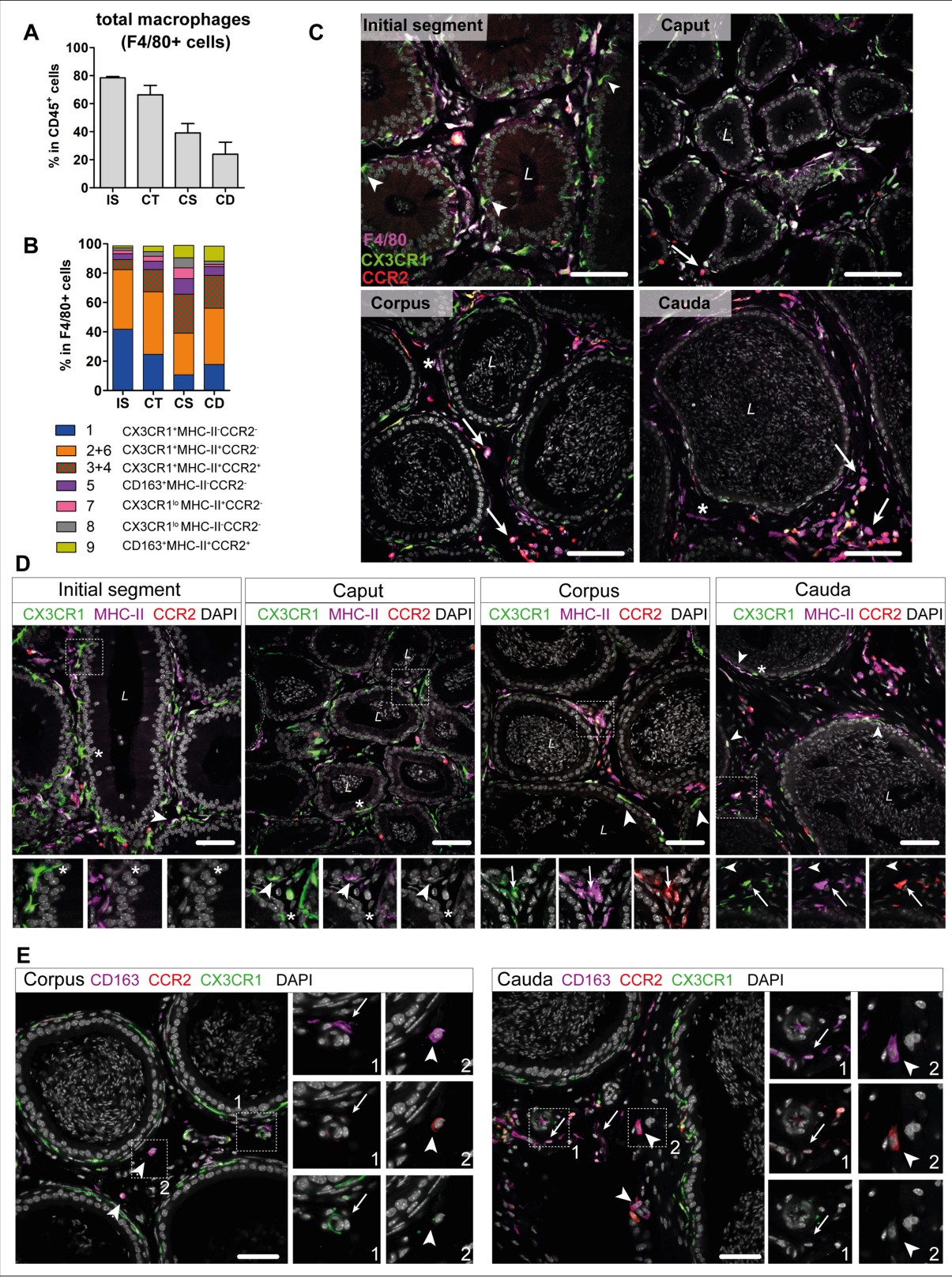

**Figure 6.** Distribution and localization of identified macrophage subgroups by flow cytometry and immunofluorescence. (**A**) Bar diagram showing the percentage of F4/80+ cells within the CD45+ population throughout the epididymal regions, assessed by flow cytometry (n=8, mean ± SD). (**B**) Stacked bar diagram displaying the percentages of identified macrophage subtypes within the F4/80+ population throughout epididymal regions assessed by flow cytometry. Markers were selected based on single-cell RNA sequencing (scRNASeq) results (n=4). (**C**) Confocal microscopy images of F4/80 staining

*Figure 6 continued on next page*

*Figure 6 continued*

(purple) on *Cx3cr1*$^{GFP}$*Ccr2*$^{RFP}$ adult reporter mice. The majority of interstitial and intraepithelial CX3CR1$^+$ cells were F4/80$^+$. Arrowheads indicate a small fraction of intraepithelial CX3CR1$^+$ F4/80 cells found within the initial segment (IS). Arrows indicate interstitial F4/80$^+$ cells that were CX3CR1$^-$ and CCR2$^+$ within caput, corpus, and cauda epididymides. Asterisks (*) label a small fraction of F4/80 single positive cells (CX3CR1$^-$CCR2$^-$) found in the corpus and cauda. Scale bar 50 μm (L=lumen). (**D**) Confocal microscopy images of MHC-II staining (purple) on *Cx3cr1*$^{GFP}$*Ccr2*$^{RFP}$ adult reporter mice. Asterisks (*) indicate intraepithelial CX3CR1$^+$MHC-II$^-$ cells within the IS and caput epididymides. Arrowheads indicate CX3CR1$^+$MHC-II$^+$ cells, lining the epididymal duct within the IS and situated within the epithelium within caput, corpus, and cauda epididymides. Arrows indicate interstitial CX3CR1$^+$MHC-II$^+$CCR2$^+$ cells additionally found within corpus and cauda epididymides. Scale bar 50 μm (L=lumen). (**E**) Confocal microscopy images of CD163 staining (purple) on *Cx3cr1*$^{GFP}$*Ccr2*$^{RFP}$ adult reporter mice in corpus and cauda epididymides. Arrows indicate CD163 single positive cells that were found in close proximity to vessels within the corpus and cauda. Arrowheads indicate CD163$^+$CCR2$^+$ cells found solitarily distributed within the interstitium in the corpus and cauda. Scale bar 50 μm.

The online version of this article includes the following figure supplement(s) for figure 6:

**Figure supplement 1.** Gating strategy of macrophage subsets according to obtained single-cell RNA sequencing (scRNASeq) data.

**Figure supplement 2.** Macrophage subpopulations within CD45+ population.

**Figure supplement 3.** Single channel reads of anti-F4/80 (purple) staining on epididymal cryo-sections from adult *Cx3cr1*$^{GFP}$*Ccr2*$^{RFP}$ reporter mice.

**Figure supplement 4.** Single channel reads of anti-MHC-II (purple) staining on epididymal cryo-sections from adult *Cx3cr1*$^{GFP}$*Ccr2*$^{RFP}$ reporter mice.

Within the blood, approximately 40–60% of Ly6C$^{hi}$ cells originated from the CD45.1 donor, indicating efficiently established chimerism. For the epididymis, all CD11b$^+$CD64$^+$ macrophages (containing all previously identified subgroups) were gated and further subdivided using CCR2 and TIMD4, respectively (*Figure 7B*, *Figure 7—figure supplement 2*), as markers for monocyte-derived and self-maintaining macrophages, as previously described in other organs (*Dick et al., 2022*). Epididymal fat was additionally investigated to examine if a directly associated neighboring tissue differs with respect to the monocyte contribution. All CCR2$^+$ cells (corresponding to clusters 3, 4, and 9), which are most abundant within the corpus and cauda (approximately 30–40 cells/mg tissue, *Figure 7D*), were exclusively of donor origin in all regions (*Figure 7C and G*). In contrast, TIMD4$^+$ cells (corresponding to cluster 5) that were found in low numbers in corpus and cauda (approximately 40–50 cells/mg tissue, *Figure 7E*) were CD45.1$^-$, indicating that this population did not originate from the donor (*Figure 7C and H*). The majority of CD64$^+$CD11b$^+$ cells were CCR2$^-$TIMD4$^-$ (double negative) and were most abundant in the IS (*Figure 7F*). This population displayed the majority of resident macrophages within the epididymis (all previously described subpopulations, mainly CX3CR1$^{hi}$ macrophages clusters 1 and 2, but also CX3CR1$^{lo}$ subpopulations clusters 7 and 8). Intriguingly, although double negative epididymal macrophages were generally less monocyte-dependent compared to macrophages located within the epididymal fat (*Figure 7C and I*), significant differences were detected between the proximal (IS, caput) and distal regions (corpus, cauda) of the epididymis. While CCR2$^-$TIMD4$^-$ macrophages within the IS and caput had only very low donor chimerism (approximately 5–10% [normalized to blood], comparable to TIMD4$^+$ macrophages), macrophages from corpus and cauda showed a much higher chimerism (30–40% [normalized to blood], *Figure 7C and I*).

These data are in line with the observed increase of CCR2$^+$ cells in the macrophage pool of corpus and cauda epididymidis upon UPEC infection. In regional terms, these data indicate a higher monocyte-dependent turnover rate of resident macrophages within the distal epididymis (corpus and cauda) in which ascending pathogens enter the epididymis first, but proposes only a minor impact of monocytes in the maintenance of macrophages within the proximal regions (IS and caput).

## Discussion

In spite of the fact that previous studies clearly showed that the different epididymal regions develop distinct immune responses following bacterial infection (*Michel et al., 2016*; *Klein et al., 2020*), the underlying mechanisms remained elusive. In our initial experiments, we confirmed findings from previous studies showing that immunopathological damage following UPEC infection occurs almost exclusively in the cauda epididymidis (*Michel et al., 2016*; *Klein et al., 2020*) with loss of epithelial integrity, interstitial fibrosis, and duct obstruction. Expanding on these previous observations, we further demonstrate that the accompanying leukocytic infiltration is characterized by a massive influx of neutrophils and monocyte-derived MHC-II$^{hi}$ macrophages with concurrent loss of epithelial

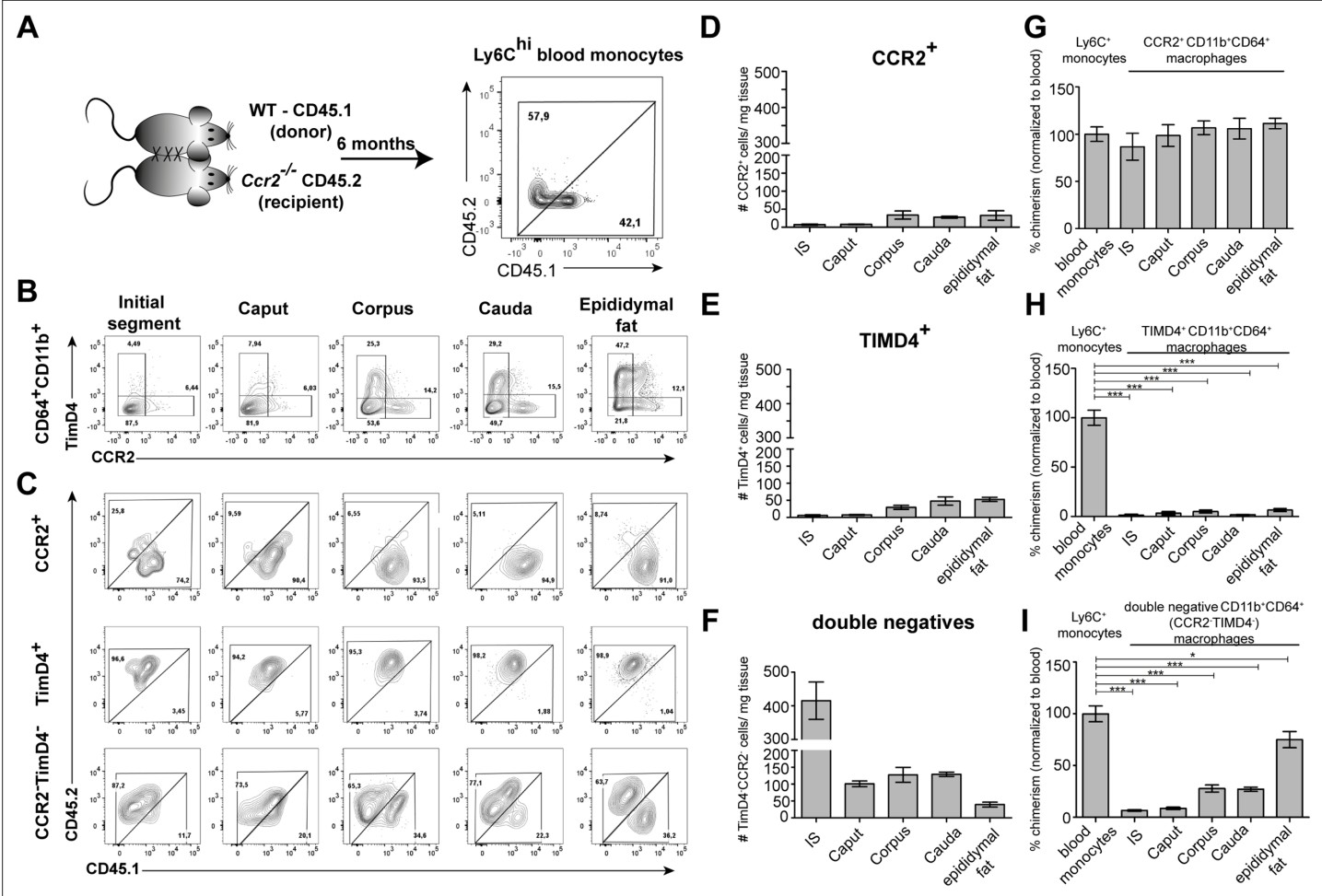

**Figure 7.** Resident macrophages differentially depend on monocytes within epididymal regions. (**A**) Parabiosis was conducted by surgically conjoining wild type CD45.1+ donor mice with CD45.2 recipient *Ccr2-/-* mice for 6 months. Donor chimerism was confirmed on CD115+CD11b+Ly6Chi monocytes. (**B**) Flow cytometry contour plots showing the segregation of resident macrophages (CD11b+CD64+) isolated from different epididymal regions using the ontogeny marker TIMD4 and CCR2. Epididymal fat served as control tissue. Plots are representative for six parabionts. (**C**) Flow cytometry contour plots showing the chimerism in CCR2+, TIMD4+, and CCR2-TimD4- macrophages within different epididymal regions based on the CD45.1 and CD45.2 labeling. Plots are representative for six parabionts. (**D–F**) Bar diagrams showing the number of CCR2+ (**D**), TIMD4+ (**E**), and CCR2-TimD4-(**F**) macrophages (CD64+CD11b+) within different epididymal regions of the analyzed recipient *Ccr2-/-* mice (mean ± SEM, n=6). (**G–I**) Bar diagrams showing the percentage of chimerism normalized to blood chimerism in CCR2+ (**G**), TimD4+ (**H**), and CCR2-TIMD4- (**I**) epididymal macrophages (CD64+CD11b+) in the recipient *Ccr2-/-* mice after 6 months (n=6, n.s.=not significant, *p<0.05, **p<.005, ***p<0.001, n=6, mean ± SEM, one-way ANOVA with Bonferroni multiple comparison test).

The online version of this article includes the following figure supplement(s) for figure 7:

**Figure supplement 1.** Gating strategy that was applied on blood samples from recipient *Ccr2-/-* mice.

**Figure supplement 2.** Representative plots (cauda) that were applied for the epididymal regions starting with general gating based on FSC and SSC in order to exclude debris and doublets.

integrity. Both neutrophils and monocyte-derived macrophages are important first defenders of the innate immune response during acute infection due to their high phagocytic activity. However, during immune response against microbes, both populations can also elicit substantial collateral tissue injury by releasing inflammatory and cytotoxic mediators that, in turn, amplify the immune response (*Segel et al., 2011*; *Kruger et al., 2015*). The role of neutrophils and monocyte-derived macrophages in tissue injury in the cauda has become evident in mice lacking CCR2, which is required for the recruitment of circulating immune cells to inflammatory sites. *Ccr2-/-* mice show a significantly reduced influx of neutrophils and inflammatory monocyte-derived macrophages concomitant with less severe tissue damage in the cauda during UPEC infection compared to wild type mice (*Wang et al., 2021*)

pointing to a role as double-edged swords in acute epididymitis by participating in both defense and inflammation-associated tissue damage. Nevertheless, bacterial virulence factors may contribute to some extent to tissue damage in the cauda following the observation that UPEC persist in this region much longer and at higher numbers than in the other parts of the organ. A further driving force of the persisting immunopathology seen in the cauda could relate to the extravasation of immunogenic spermatozoa through the damaged epithelial barrier, which may trigger accumulated MHC-II$^{hi}$ macrophages and lymphocytes (B and T cells) toward an adaptive immune response against spermatozoal neo-antigens. This is supported by an increase of B and T cell populations and upregulation of gene sets associated with their activation as well as granuloma formation, indicating a transition from innate to adaptive immune response limited to the cauda. Of note, granulomas can be induced by interstitial sperm injection alone also leading to massive tissue destruction in the cauda epididymidis (*Itoh et al., 1999*) and formation of anti-sperm antibodies as seen in another model of *E. coli*-elicited epididymitis and in epididymitis patients (*Ingerslev et al., 1986*; *Nashan et al., 1993*; *Lotti et al., 2018*; *Silva et al., 2021*).

Contrasting to the strong pro-inflammatory processes within the cauda, the caput remains mostly unaffected – an observation that previously raised the question whether the caput is either non-responsive or to a lesser extent responsive compared to the cauda. As bacteria are initially present in the caput, albeit at lower numbers and for a shorter time (potentially due to a faster bacterial clearance potential as evidenced by the *ex vivo* approach), it can be excluded that the lower bacterial load is an explanation for the differential immune response in the caput. The very mild and transient immune response in the caput is characterized by the upregulation of a very small number of genes that are indicative of a limited inflammatory response triggered by the pathogens, such as the alarmins *S100a8* and *S100a9*. Interestingly, both alarmins have previously been demonstrated to be upregulated within the kidney and bladder during UPEC-elicited urinary tract infection, but did not substantially contribute to an effective host immune response (*Dessing et al., 2010*). Possibly, the upregulation of *S100a9* within the caput could drive macrophages to polarize to an anti-inflammatory and immunosuppressive phenotype as seen in the testis following UPEC infection (*Fan et al., 2021*). A clear indication for a regionalized immune response with a predominant reaction in the cauda is derived from this and other studies that use an inflammatory stimuli such as LPS that act simultaneously on all regions in vivo and in vitro rather than gradually ascending such as an in vivo UPEC infection (*Figure 2—figure supplement 3*, *Silva et al., 2018*; *Wang et al., 2019*; *Wijayarathna et al., 2020*).

Taken together, it is evident that fundamental differences must exist in the immunological milieus of the epididymal regions that most likely rely on leukocyte subpopulations that gradually change in phenotype throughout the organ. Initial evidence came from previous studies (*Da Silva et al., 2011*; *Da Silva and Smith, 2015*; *Da Silva and Barton, 2016*; *Voisin et al., 2018*; *Battistone et al., 2020*; *Mendelsohn et al., 2020*). However, the full extent of the heterogeneity of resident immune cells and their identity remained elusive. Our data unravel the transcriptional identity and tissue location of extravascular immune cells and further support the existence of distinct immunological environments along the epididymal duct that are tailored to the respective needs of the microenvironment.

Overall, macrophages constitute the major immune cell population, especially within the IS. Along the epididymal duct macrophages exhibit a dense network consisting of several transcriptional distinct subpopulations that populate different niches according to their homeostatic, reparative, and inflammatory properties. CX3CR1$^{hi}$ macrophages that possess a homeostatic and sensing profile are situated within and around the epididymal epithelium with highest abundance in the IS. Data about the density of these cells in the epididymal regions, however, differ among studies, probably due to methodological differences (see 'Public Review' for details, *Voisin et al., 2018*; *Battistone et al., 2020*). The morphological abundance of these macrophages, particularly the exhibition of long dendrites toward the lumen, is consistent with previous observations (*Da Silva et al., 2011*; *Battistone et al., 2020*). The transcriptional profile of intraepithelial CX3CR1$^{+}$ cells combined with the known high phagocytic potential toward apoptotic epithelial cells (i.e. within the IS, *Smith et al., 2014*) and pathogens (*Battistone et al., 2020*) indicates a central function in tissue homeostasis and immune surveillance in order to efficiently maintain epithelial integrity that in turn is mandatory for maintaining the luminal microenvironment required for proper sperm maturation. Of note, the high density of sensing CX3CR1$^{hi}$ macrophages in combination with the narrow lumen of the IS indicates a potential function of this region as 'immunological bottleneck' in which the luminal content

is constantly monitored in order to induce tolerance toward immunogenic sperm antigens and to eliminate pathogens from further ascend to the testis. Whether these immune cells directly influence sperm maturation processes needs to be elucidated. The presence of CX3CR1$^+$ cells within the IS was previously described, however, were initially related to dendritic cells (*Da Silva et al., 2011*) and subsequently generally as MP based on morphology and partial CD11c expression (*Da Silva and Smith, 2015*; *Da Silva and Barton, 2016*). In our study the transcriptional profile clearly indicates a macrophage phenotype with sensing functions.

In contrast, the distal regions (corpus, cauda) are populated by a more heterogeneous macrophage pool consisting of less intraepithelial CX3CR1$^+$ macrophages, but higher abundance of interstitial pro-inflammatory monocyte-derived CCR2$^+$MHC-II$^+$, vasculature-associated TLF$^+$ macrophages (expressing a combination of *Timd4, Lyve1, Folr2*; marker used in the present study CD163) as well as CX3CR1$^-$TLF$^-$CCR2$^-$ macrophages (contained in clusters 7 and 8). The co-existence of these three populations was recently reported to be conserved across organs (*Dick et al., 2022*) and, similar to other organs, the different macrophage pools have distinct monocyte contributions. While TLF$^+$ macrophages are rather self-renewing and CCR2$^+$ macrophages monocyte-dependent, TLF$^-$CCR2$^-$ macrophages (including CX3CR1$^+$ macrophages in our parabiosis experiment with alternating MHC-II levels) are partially dependent on monocytes in distal, but not in proximal epididymal regions. These findings support the conclusion that local environmental factors could influence parameters that regulate monocyte entry and replacement of distinct macrophage populations in different regions of the epididymis.

The co-existence of antigen-presenting myeloid cell populations (macrophages, monocytes, dendritic cells) with lymphocyte subtypes (NK cells, αβ T cells, γδ T cells, B cells) within the distal regions of the epididymis implies an environment of immune responsiveness. As we found a higher proportion of monocytes-derived cells and conventional DC 1 and 2 (including a small fraction of activated *Ccr7$^+$* DC) within the distal regions, an ongoing antigen sampling and interaction with the draining lymph node can be assumed but would require further confirmation before a better understanding of the region-specific role of migratory myeloid cells in the epididymal immune regulation can be achieved. The existence of intraepithelial and interstitial lymphocyte subpopulations with innate-like characteristics (i.e. NK cells, γδ T cells), predominantly within the distal regions, implies a contribution of these cells to the onset of immune responses against pathogens. Both populations generally function as key responders to barrier stress signals in mucosal tissues and accelerate pro-inflammatory processes by secreting effector cytokines, particularly IL-17- and IFNγ (*Shi et al., 2011*; *Papotto et al., 2017*). Since the function of innate lymphoid cells highly depends on their activation status and functional polarization within the periphery (*Bonneville et al., 2010*; *Klose and Artis, 2020*), local environmental factors may determine regulatory vs. cytotoxic functions of NK and γδ T cells in different epididymal regions.

As a limitation, our scRNASeq approach supplies 'only' a snapshot of extravascular immune cells within the epididymis at a defined time (in the adult) neglecting times of residency for, for example, myeloid and lymphoid immune cell populations that possess migratory capabilities and patrol between non-lymphoid tissues and draining lymph nodes, a process required for immune surveillance and induction of immune responses or tolerance (*Hampton and Chtanova, 2019*).

Together, our data provide the first atlas of extravascular CD45$^+$ cells within the murine epididymis under normal conditions. Strategic positioning of identified immune cell populations strongly indicates the existence of distinct immunological landscapes at the opposing ends of the epididymal duct that, in turn, is considered as a main driver for the observed differences in the intensity of immune responsiveness upon bacterial infection. We believe that the data in the present study provide a valuable starting point and common research platform for future studies on the organ-specific function of these populations in epididymal immunity.

## Methods
### Mice

All mice used in this study were purchased from Charles River and Jackson Laboratories and/or bred under pathogen-free conditions prior to use at the animal facilities of Justus Liebig-University Giessen, Germany (C57BL/6J wild type [Charles River], *Cx3Cr1*$^{GFP}$*Ccr2*$^{RFP}$ [JAX ID: 032127, Jackson

Laboratories]), The Toronto General Research Institute, Canada (C57BL/6J CD45.1 [JAX ID: 002014], *Ccr2*[-/-] [JAX ID: 004999]) and the Central Animal Facility at Hannover Medical School (Tcrd-H2BeGFP, JAX ID: 016941).

All animal experiments were approved by the respective local Animal Ethic Committees (Germany: Regierungspräsidium Giessen GI20/25 No. G60/2017, GI20/25 No. G71/2019, the Nds. Landesamt für Verbraucherschutz und Lebensmittelsicherheit 2017/141 and 2021/276, as well as Canada AUP: 4054.37). Killing of wild type C57BL/6J and *Cx3Cr1*[GFP]*Ccr2*[RFP] mice without any prior treatment had been declared to the Animal Welfare Officer of Justus-Liebig-University Giessen, Germany (Registration No. M_684 and M_ 755, respectively). Experiments were conducted in strict accordance with the Guidelines of the Care and Use of Laboratory Animals of the German law for animal welfare, the European legislation for the protection of animals for scientific purposes (2010/63/EU) and the Guidelines of the Canadian Council of Animal Care. For euthanasia prior to organ collection, mice were deeply anesthetized by inhalation of 4–5% isoflurane followed by cervical dislocation, if not otherwise stated.

## Induction of acute bacterial epididymitis in mice

UPEC strain CFT073 (characterized by *Welch et al., 2002*) were provided by the Institute of Medical Microbiology, Justus-Liebig-University Giessen, Germany, and cultured as described previously (*Bhushan et al., 2008*). To elicit an ascending canalicular infection, vasa deferentia were bilaterally ligated followed by an intravasal injection of UPEC (in sterile 0.9% NaCl) close to the cauda (5 μl containing $1 \times 10^5$ colony forming units [CFU]) using a Hamilton syringe. Control 'sham' mice underwent the same surgical procedure with an intravasal injection of 5 μl sterile 0.9% NaCl. Mice were sacrificed at days 1, 3, 5, 7, 10, 14, and 21 after infection by isoflurane narcosis and cervical dislocation. For each time point, three to six mice were used per experimental approach. For all subsequent approaches, at least two independent experiments were conducted.

## RNA extraction, RNASeq, and whole transcriptome analysis

RNA was extracted from caput (segment 1–5, including the IS), corpus (segment 6–7), and cauda (segment 8–10) samples using QIAzol Lysis Reagent (Qiagen) following the manufacturer's recommendation using a bead-based tissue homogenizer (Retsch, using 2.8 mm stainless steel beads). RNA was purified using the RNeasy Mini Kit (Qiagen) with an on-column DNase digestion using RNAse-free DNase Set (Qiagen) for 30 min to eliminate genomic DNA contamination. Total RNA and library integrity was verified with LabChip Gx Touch 24 (Perkin Elmer, MA, USA). Ten ng total RNA was used as template for SMARTer Stranded Total RNA-Seq Kit – Pico Input Mammalian (Takara Bio) following the manufacturer's recommendation.

Sequencing was conducted on the NextSeq500 instrument (Illumina, CA, USA) using v2 chemistry with 1×75 bp single end setup. The resulting raw reads were assessed for quality, adapter content, and duplication rates with FastQC (*Andrews, 2010*). Trimmomatic version 0.39 was employed in order to trim reads after a quality drop-down below a mean of Q20 in a window of 5 nucleotides (GRCm38.p5) using STAR 2.6.1d with the parameter '—outFilterMismatchNoverLmax 0.1' to increase the maximum ratio of mismatches to mapped length to 10% (*Dobin et al., 2013*). The number of reads aligning to genes was counted with the featureCounts 1.6.5 tool from the Subread package (*Liao et al., 2013*). Only reads mapping at least partially inside exons were admitted and aggregated per gene. Reads overlapping multiple genes or aligning to multiple regions were excluded. DEG were identified using DESeq2 version 1.18.1 (*Love et al., 2014*). Only genes with a minimum fold change of ±1.5 (log2±0.59), a maximum Benjamini-Hochberg corrected p-value of 0.05, and a minimum combined mean of 5 reads were considered to be significantly differentially expressed. The Ensembl annotation was enriched with UniProt data (release 06.06.2014) based on Ensembl gene identifiers (*UniProt Consortium, 2014.*)

## Determination of CFU

For each time point (1, 3, 5, 7, and 10 days p.i.), four biological replicates were used to assess the bacterial loads in the different epididymal regions. Data were obtained from two independent experiments. Tissue was collected and separated under sterile conditions in the IS, caput, corpus, and cauda before homogenization in 250 μl sterile ice-cold PBS. Tenfold serial dilutions were prepared and spread onto Luria broth (LB) agar plates (10 mg/ml tryptone, 5 mg/ml yeast extract, 10 mg/

ml NaCl, and 15 mg/ml agar agar [pH 7.0]). Plates were incubated upside-down at 37°C overnight before CFU were counted and calculated in relation to the previously determined tissue weight (per mg of used tissue). Pure *E. coli* were plated as positive control, whereas PBS only that was kept in similar tubes as the samples prior to plating to exclude contaminations within PBS solution and used tubes.

## Histological staining (modified Masson-Goldner trichrome staining)

Bouin's-fixed (5 hr) and paraffin-embedded epididymides were cut into 5 μm sections. Deparaffinized and rehydrated tissues were stained for 2 min with Weigert's iron hematoxylin for nuclear labeling (1:1 mixture of stock solution I [10 mg/ml hematoxylin 96% ethanol] and stock solution II [11.6 mg/ml $FeCl_3$ in 2.5% HCl]) followed by blueing in running tap water for 15 min. Sections were rinsed in 1% acetic acid followed by cytoplasmic staining with Ponceau-Acid Fuchsin (10 mg/ml Ponceau de Xylidine, 5 mg/ml Acid Fuchsin in 2% acetic acid) for 5 min. Sections were rinsed in 1% acetic acid for 3 min before incubated in 5% phosphotungstic acid for 30 min (under visual control). Sections were rinsed in distilled water for 3 min, followed by staining of connective tissue using aniline blue – orange G solution (5 mg/ml aniline blue and 20 mg/ml orange G in 8% acetic acid for 30 min). Subsequently, sections were rinsed in 1% acetic acid followed by dehydration in increasing concentrations of ethanol and xylene and mounting using Entellan (Sigma-Aldrich). Images were acquired using a Leica DM750 microscope (Leica Microsystems, Wetzlar, Germany). In order to create whole organ images, single-captured images were composed using Inkscape V0.92.4. Morphometric analyses were performed using ImageJ V1.53a.

In order to assess the luminal diameter in different epididymal regions, 20–30 duct cross sections were measured per segment from the opposite apical surfaces of the ductal epithelium using ImageJ V1.53a. Measurements from segment 1–5 were averaged and summarized as 'caput' (incl. IS). Measurements from segment 8–10 were averaged and summarized as 'cauda'. In total, three biological replicates were used per infection time point and experimental group. Area of immune cell infiltrates/granuloma area was measured on sections stained with Masson-Goldner trichrome staining using ImageJ V1.53a. For each biological replicate, four to five sections were measured and averaged.

## Disease score of acute bacterial epididymitis

A disease scoring system was slightly modified from a previously reported disease score established for experimental autoimmune epididymo-orchitis (*Wijayarathna et al., 2020*), in order to categorize and compare the observed histopathological alterations throughout the time course. The scoring system considered the following aspects:

0 – No histological alterations, normal tissue architecture.
1 – Scattered/focal mild histological alterations.
2 – Mild histological alterations (mild reduction of the luminal diameter, mild interstitial fibrosis).
3 – Mild to moderate histological alterations (mild interstitial fibrosis, moderate luminal diameter reduction, focal and mild epithelial damage).
4 – Moderate histological alterations (moderate interstitial fibrosis, moderate luminal diameter reduction, moderate epithelial damage).
5 – Severe histological alterations (severe interstitial fibrosis, severe luminal diameter reduction, loss of epithelial integrity, presence of 'ductal ghosts').

## *Ex vivo* organ culture and cytokine measurement

Epididymides were isolated from 10- to 12-week-old C57BL/6J mice and separated into IS (segment 1), caput (segment 2–5), corpus (segment 6–7), and cauda (segment 8–10, *Figure 1—figure supplement 2* n=4). Organ pieces were transferred into a 24-well plate containing RPMI media only and then pre-incubated for 15 min at 34°C with 5% $CO_2$ before 50 ng/ml LPS was added. After 6 hr incubation, supernatants and organ pieces were collected. Protein concentrations of inflammatory cytokines were determined by LegendPlex Multiplex Assay (BioLegend) using the predefined mouse inflammation panel according to the manufacturer's instructions. Cytokine levels were determined in both tissue homogenate (protein extraction using RIPA-buffer and quantification using Bradford Assay) and the supernatant, producing similar results.

## Gentamicin assay

Epididymides were isolated from 10- to 12-week-old C57BL/6J mice and cultured *ex vivo* as described above (see '*Ex vivo* organ culture and cytokine measurements'). $1 \times 10^6$ UPEC were added to the organ culture and incubated for 4 hr. Supernatants were carefully removed and organ pieces were washed twice with sterile PBS. Subsequently, organ pieces were incubated within 1 ml RPMI media containing 200 µg/ml gentamicin for 1 hr at 34°C and 5% $CO_2$ resulting in elimination of extracellular bacteria while intracellular bacteria were unaffected. Organ pieces were washed twice with PBS prior to tissue homogenization in 250 µl sterile ice-cold PBS. Homogenates were spread onto LB agar plates and incubated upside-down 24 hr at 37°C. Colonies were counted and calculated in relation to the previously determined tissue weight (per 10 mg of used tissue). Data were obtained from two independent experiments. UPEC alone were plated as positive control. A bacterial suspension that was treated with gentamicin under the same conditions as the organ pieces showed no colony forming as proof of antibiotic effectiveness.

## Cell preparation and surface staining for flow cytometry

For flow cytometric analyses, mice were sacrificed by deep isoflurane anesthesia and cervical dislocation. In order to eliminate the majority of intravascular CD45$^+$ cells, mice were perfused with PBS by inserting a 30 G needle into the left ventricle of the heart, while the right ventricle was opened by a small incision. Up to 50 ml PBS were carefully and continuously injected for 5–10 min until the tissue in the scrotal area cleared (especially the highly vascularized IS). For flow cytometric analyses of UPEC-infected mice, epididymides were separated into caput (containing the IS) and cauda and single organs were used for single-cell suspension (due to individual differences in immune responses). For flow cytometric analyses under physiological conditions, epididymides were dissected into IS, caput, corpus, and cauda and the tissue of three mice were pooled due to the small tissue size (5–7 mg per organ piece) in order to obtain sufficient numbers of cells. Collected tissue was mechanically dissociated by chopping followed by enzymatic digestion for 45 min at 37°C in DMEM containing collagenase D (1.5 mg/ml, Roche) and DNase I (60 U/ml, Sigma). Digested suspensions were aspirated through 30 G needles four to six times and filtered through a 70 µm cell strainer before centrifugation at $400 \times g$ for 10 min at 4°C. Single-cell suspensions were incubated with red blood cell lysis (RBC lysis buffer, Qiagen) for 3 min before pelleted by centrifugation ($400 \times g$, 10 min at 4°C). Cells were re-suspended in PBS and stained with a fixable viability dye in order to assess viability (different viability dyes were used depending on the respective panel: ZombieAqua [Biolegend, 423101], ZombieNIR [BioLegend, 423105], Viobility 405/452 Fixable Dye [Miltenyi, 103-109-816]) following the respective manufacturer's recommendation. To block nonspecific binding of antibodies to mouse cells expressing Fc receptors, cell suspensions were incubated with Fc blocking reagent (Miltenyi, 130-092-575) following the manufacturer's recommendations. Cells were stained with antibodies listed in the Appendix 1—Key resources table for 30 min at 4°C in 50 µl MACS Quant buffer (2 mM EDTA and 0.5% BSA in PBS). Respective controls were used by omitting the target antibody and incubating with the respective isotype control under the same conditions. For the infection analysis, cells were fixed with 4% PFA for 15 min at room temperature. Cells were washed twice with MACS Quant buffer before re-suspension in 200–500 µl MACS Quant buffer (depending on cell numbers). Flow cytometry was performed using a MACSQuant Analyzer 10 (steady-state analysis) and BD LSRFortessa Cell Analyzer (for infection analysis). All obtained data were analyzed with FlowJo software version 10.8.1. Graphs were generated using GraphPad Prism 5.

## Gating strategy and panel constellation for flow cytometry

A general gating strategy (outlined in *Figure 4—figure supplement 1*) was performed on each sample prior to the gating based on surface staining. Briefly, debris and sperm were excluded by SSC-A/FSC-A followed by two-step single-cell gating (FSC-H/FSC-A and SSC-H/SSC-A) and Live/Dead discrimination using viability dyes listed in the Appendix 1-key resource table.

## Gating of immune cells in UPEC-infected and sham mice (Figure 2—figure supplement 1)

Panel: ZombieNIR, PerCP/Cy5.5, GR1-BV711, NK1.1-BV605, B220-BV510, F4/80-PE/Dazzle594, CX3CR1-PE/Cy7, CD11c-BV650, MHC-II-BV786, CCR2-FITC, CD3-AF700.

Neutrophils:

> GR-1$^+$SSC$^{hi}$

Monocytes:

> GR-1$^+$SSC$^{lo}$

B cells:
GR-1$^-$CD45R/B220$^+$
NK cells:
GR-1$^-$ CD45R/B220$^-$NK1.1$^+$
Macrophages:
GR-1$^-$CD45R/B220$^-$NK1.1$^-$F4/80$^+$

> *distinguished between* CX3CR1$^+$ and CX3CR1$^-$ and CCR2$^+$

Total DC (containing cDC1 and cDC2):

> GR-1$^-$CD45R/B220$^-$NK1.1$^-$F4/80$^-$CD11c$^+$MHC-II$^+$

T cells:

> GR-1$^-$CD45R/B220$^-$NK1.1$^-$F4/80$^-$CD11c$^+$MHC-II$^+$CD3$^+$

## Gating of immune cells under physiological conditions

### Dendritic cells

Panel: F4/80-BV421, Clec9a-BV510, CD45-AF488, CD209a-PE, CD11b-PerCP/Cy5.5, MHC-II-APC, ZombieNIR Fixable Dye – see 'Dendritic cell steady-state panel' in Appendix 1—key resources tablefor more details.

> *Conventional dendritic cells 1 (cDC1):* CD45$^+$F4/80$^-$MHC-II$^{hi}$Clec9a$^+$
> *Conventional dendritic cells 2 (cDC2):* CD45$^+$F4/80-MHC-II$^{hi}$CD209a$^+$

### Lymphocytes

Panel: NK1.1-BV421, ZombieAqua Fixable Dye, CD3-FITC, B220 (CD45R)-PE, TCRbeta-PE/Cy7, TCRgd-APC, CD45-APC/Fire750 – see 'Lymphocyte steady-state panel' in Appendix 1—key resources table for more details.

> *B cells*: CD45$^+$CD3$^-$B220$^+$
> *T cells:* CD45$^+$B220$^-$CD3$^+$ NK1.1$^-$
> *αβ T cells:* CD45$^+$B220$^-$CD3$^+$ NK1.1$^-$TCRbeta$^+$TCRγδ$^-$
> *γδ T cells:* CD45$^+$B220$^-$CD3$^+$ NK1.1$^-$TCRbeta$^-$TCRγδ$^+$
> *NK cells:* CD45$^+$B220$^-$CD3$^-$NK1.1$^+$

### Macrophage subpopulation steady state

Panel: CX3CR1-BV421, ZombieAqua Fixable Dye, CCR2-FITC, CD45-PerCP/Cy5.5, F4/80-PE/Cy7, CD163-APC, MHC-II-APC/Cy7 – see 'Macrophages steady-state panel' in Appendix 1—key resources tablefor more details.

> Cluster 1: F4/80$^+$CX3CR1$^{hi}$CCR2$^-$MHC-II$^-$
> Cluster 2+6: F4/80$^+$CX3CR1$^{hi}$CCR2$^-$MHC-II$^+$
> Cluster 3+4: F4/80$^+$CX3CR1$^+$CCR2$^+$MHC-II$^+$
> Cluster 5: F4/80$^+$CD163$^+$CCR2$^-$
> Cluster 7: F4/80$^+$CX3CR1$^{lo}$CCR2$^-$MHC-II$^+$
> Cluster 8: F4/80$^+$CX3CR1$^{lo}$CCR2$^-$MHC-II$^-$
> Cluster 9: F4/80$^+$CD163$^+$CCR2$^+$MHC-II$^+$

## Monocytes

CD45+Ly6G-Ly6C+CD11b^hi

## Parabiosis

Male donor mice (B6 CD45.1, JAX ID: 002014, *Janowska-Wieczorek et al., 2001*; *Schluns et al., 2002*; *Yang et al., 2002*) and recipient mice (CD45.2 *Ccr2*^-/- JAX ID: 004999, *Boring et al., 1997*) were laterally shaved and conjoined by matching skin incisions from behind the ear to the tail as described previously (*Dick et al., 2019*). Six months after parabiosis surgery, mice were sacrificed by $CO_2$ inhalation prior to blood and organ collection. In total, six recipient mice were analyzed. Cells were isolated from the four main epididymal regions (IS, caput, corpus, cauda) and epididymal fat for flow cytometry as outlined above. The chimerism for each macrophage subpopulation was normalized to blood monocytes in the recipient mouse (% normalized chimerism = (%donor cells in recipient/%Ly6C^hi monocyte donor cells in recipient) *100) according to *Dick et al., 2019*.

Gating strategy (as outlined in *Figure 7—figure supplements 1 and 2*): Debris and sperm were excluded by SSC-A/FSC-A followed by a two-step doublet exclusion based on FSC-H/FSC-A and SSC-H/SSC-A. Total resident macrophages from all epididymal regions and epididymal fat were identified as CD45+CD11b+CD64+. TIMD4+ macrophages and CCR2+ macrophages were gated as internal controls for self-renewing and monocyte-derived macrophages, respectively (according to *Dick et al., 2022*). CD45+CD11b+CD64+CCR2-TimD4- macrophages represented the entirety of all resident macrophage subpopulations. Blood monocytes were identified as CD45+Ly6G-CD115+CD11b+Ly6C+.

## Single-cell preparation of extravascular CD45+ cells

Ten 10-week-old male wild type C57BL/6J mice were intravenously injected with an APC/Cyanine 7-conjugated anti-mouse CD45.2 antibody (Clone 104, BioLegend 109824, RRID: AB_830789) 5 min prior to euthanasia by $CO_2$ inhalation. Single-cell suspensions of epididymal regions (IS, caput, corpus, cauda) were prepared as previously described (see 'Cell preparation and surface staining for flow cytometry'), with inclusion of 60 U/ml hyaluronidase type I-S (Sigma, H3506) in the digestion buffer. Cells were stained with a PerCP-Cyanine 5.5-conjugated CD45.1 antibody (BioLegend 110728, RRID: AB_893346). Stained single-cell suspensions of all mice were pooled to obtain enough immune cells for sorting. Single live CD45.1+CD45.2- immune cells were sorted on the BD Aria Fusion (BD Bioscience) for scRNASeq.

## Library preparation and data analysis

Single-cell suspensions were prepared as outlined previously (*Dick et al., 2019*; *Dick et al., 2022*) using the 10× Genomics Single Cell 3' v3 Reagent Kit user guide based on individually calculated cDNA concentrations. Briefly, cell suspensions were washed twice with PBS supplemented with 0.04% BSA and centrifuged at 330 × *g* for 6 min. The appropriate volume for droplet generation was assessed by counting live cells using Tryptan Blue staining and a hemocytometer. Reverse transcription was performed in a pre-chilled 96-well plate (heat-sealed) using a Veriti 96-well thermal cycler (Thermo Fisher). cDNA was recovered using 10x-associated Recovery Agent followed by amplification and purification using SPRIselect beads (Beckman) following the manufacturer's recommendation. After diluting samples in a 4:1 ratio (elution buffer [Qiagen]:cDNA), cDNA concentration was determined using a Bioanalyzer (Agilent Technologies).

Sequencing libraries were produced by loading samples on the 10× Chromium. Generated libraries were processed as recommended by the methods provided from 10× Genomics. Expression matrices were generated using Cell Ranger (10× Genomics). Obtained raw base call (BCL) files from the HiSeq2500 sequencer were demultiplexed into FastQ files. Sequencing reads were aligned to the mouse genome/transcriptome (mm10) and counted by StarSolo. After library preparation and cell mapping (StarSolo), 13,015 data points were identified as valid cells (2076 within IS, 3791 within caput, 4523 within corpus, 2625 within cauda). Preprocessed counts were further analyzed using Scanpy. Basic cell quality control was conducted by taking the number of detected genes and mitochondrial content into consideration. In total, 49 cells that did not express at least 300 genes or had a mitochondrial content greater than 10%, were removed. Genes were filtered out if they were detected in less than 30 cells (<0.2%). Raw counts per cell were normalized to the median count over

all cells and transformed into log space to stabilize variance. Dimensionality reduction was performed by PCA, retaining 50 principal components. Subsequent steps, for example, low-dimensional UMAP embedding and cell clustering via community detection, were based on the initial PCA. Final data visualization was performed using the Scanpy and CellxGene package.

## Immunofluorescence

Epididymides from *Cx3Cr1*^GFP^*Ccr2*^RFP^ reporter mice (JAX ID: 032127, *Jung et al., 2000*; *Saederup et al., 2010*), Tcrd-H2BEGFP (JAX ID: 016941, *Prinz et al., 2006*), and C57BL/6J mice (Charles River) were fixed with ROTIHistofix 4% (Carl Roth, Germany) for 5 hr followed by washing in phosphate buffer and incubation in 30% sucrose overnight at 4°C before embedding in OCT media and storage at –80°C. Twenty µm cryo-sections were prepared using a Leica Cryotome CM1850 and air-dried for 20 min followed by a 20 min post-fixation in 100% methanol at –20°C. After washing in TBS-T (TBS+0.05% Tween, pH 7.6), sections were permeabilized using 0.2% Triton-X-100 in TBS-T for 20 min at room temperature. Washed sections were incubated for 30 min in blocking solution (3% BSA, 10% normal goat serum in TBS-T). Primary antibodies (for further specification, see Key resources table, Appendix 1—key resources table) were diluted in blocking solution (F4/80 [Bio-Rad]: 10 µg/ml, Ly6G [abcam]: 1 µg/ml, MHC-II [BioLegend]: 5 µg/ml, CD163 [Invitrogen]: 5 µg/ml, LY6C [BioLegend]: 5 µg/ml, CD3 [BioLegend]: 10 µg/ml, Clec9a [R&D Systems]: 15 µg/ml, NCR1 [abcam]: 7 µg/ml, CD19 [abcam]: 8 µg/ml, DC-Sign [Santa-Cruz]: 10 µg/ml) and incubated overnight at 4°C. Secondary antibodies were diluted in TBS-T according to the manufacturer's recommendation and incubated 1 hr in a dark chamber at room temperature. Sections were thoroughly washed four times for 10 min in TBS-T before mounting with Invitrogen ProLong Gold Antifade Mountant with DAPI (Thermo Fisher). Sections were imaged with a Zeiss LSM 710 confocal microscope and Zen Software version 14.0.26.201.

## Acknowledgements

The study was supported by grants from the Deutsche Forschungsgemeinschaft (DFG), Monash University, and the Medical Faculty of Justus-Liebig University to the International Research Training Group on 'Molecular pathogenesis of male reproductive disorders' (GRK 1871, AM), as well as the von Behring-Roentgen Stiftung (CP). We would like to thank Martina Hudel for preparing the UPEC suspensions and Suada Fröhlich for assistance in tissue processing.

---

# Additional information

### Funding

| Funder | Grant reference number | Author |
| --- | --- | --- |
| Deutsche Forschungsgemeinschaft | GRK 1871 | Andreas Meinhardt |
| von Behring-Roentgen Stiftung | 69-0029 | Christiane Pleuger |

The funders had no role in study design, data collection and interpretation, or the decision to submit the work for publication.

---

### Author contributions

Christiane Pleuger, Conceptualization, Data curation, Formal analysis, Supervision, Validation, Investigation, Visualization, Methodology, Writing - original draft, Project administration, Writing – review and editing; Dingding Ai, Minea L Hoppe, Laura T Winter, Daniel Bohnert, Dominik Karl, Crystal Kantores, Validation, Investigation, Visualization, Methodology; Stefan Guenther, Resources, Software, Formal analysis, Validation, Investigation; Slava Epelman, Validation, Investigation, Methodology, Writing – review and editing; Monika Fijak, Investigation, Methodology; Sarina Ravens, Resources, Validation, Investigation; Ralf Middendorff, Kate L Loveland, Supervision, Project administration; Johannes U Mayer, Resources, Investigation; Mark Hedger, Supervision, Project administration, Writing – review

and editing; Sudhanshu Bhushan, Validation, Investigation, Project administration; Andreas Meinhardt, Resources, Supervision, Funding acquisition, Project administration, Writing – review and editing

### Author ORCIDs
Christiane Pleuger (iD) http://orcid.org/0000-0002-2807-7048
Johannes U Mayer (iD) http://orcid.org/0000-0001-6225-7803
Sudhanshu Bhushan (iD) http://orcid.org/0000-0002-9088-8108

### Ethics
All animal experiments were approved by the respective local Animal Ethic Committees (Germany: Regierungspräsidium Giessen GI20/25 G60/2017, GI20/25 G71/2019, the Nds. Landesamt für Verbraucherschutz und Lebensmittelsicherheit 2017/141 and 2021/276, as well as Canada AUP: 4054.37). Killing of wild type C57BL/6J and Cx3Cr1GFPCcr2RFP mice without any prior treatment had been declared to the Animal Welfare Officer of Justus-Liebig-University Giessen, Germany (Registration No. M_684 and M_ 755, respectively). Experiments were conducted in strict accordance with the Guidelines of the Care and Use of Laboratory Animals of the German law for animal welfare, the European legislation for the protection of animals for scientific purposes (2010/63/EU) and the Guidelines of the Canadian Council of Animal Care. For euthanasia prior to organ collection, mice were deeply anesthetized by inhalation of 4-5 % isoflurane followed by cervical dislocation, if not otherwise stated.

### Decision letter and Author response
Decision letter https://doi.org/10.7554/eLife.82193.sa1
Author response https://doi.org/10.7554/eLife.82193.sa2

## Additional files

### Supplementary files
• MDAR checklist

### Data availability
Sequencing data have been deposited and are publicly available in GEO under accession code GSE208244.

The following dataset was generated:

| Author(s) | Year | Dataset title | Dataset URL | Database and Identifier |
|---|---|---|---|---|
| Pleuger C, Guenther S, Epelman S, Kantores C, Bhushan S, Meinhardt A | 2022 | The regional distribution of resident immune cells shapes distinct immunological environments along the murine epididymis | https://www.ncbi.nlm.nih.gov/geo/query/acc.cgi?acc=GSE208244 | NCBI Gene Expression Omnibus, GSE208244 |

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

# Appendix 1

## Appendix 1—key resources table

| Reagent type (species) or resource | Designation | Source or reference | Identifiers | Additional information |
|---|---|---|---|---|
| Strain, strain background (*Escherichia coli*) | CFT073 | *Welch et al., 2002* | NCBI: txid19.9310 | Provided by T.Chakraborty, Justus-Liebig-University, Giessen, Germany |
| Strain, strain background (*Mus musculus, male*) | C57BL/6 J wild type | Charles River | JAX ID: 000664 | 10–12 weeks old |
| Strain, strain background (*Mus musculus, male*) | B6.129(Cg)-*Cx3cr1*[tm1Litt] *Ccr2*[tm2.1Ifc]/JernJ (*Cx3cr1*[GFP]*Ccr2*[RFP]) | Jackson Laboratory *Jung et al., 2000*; *Saederup et al., 2010* | JAX ID: 032127 | |
| Strain, strain background (*Mus musculus, male*) | B6.SJL-*Ptprc*[a] *Pepc*[b]/BoyJ (B6 CD45.1) | Jackson Laboratory (*Janowska-Wieczorek et al., 2001*; *Schluns et al., 2002*; *Yang et al., 2002*) | JAX ID: 002014 | |
| Strain, strain background (*Mus musculus, male*) | B6.129S4-*Ccr2*[tm1Ifc]/J (*Ccr2*[-/-]) | Jackson Laboratory (*Boring et al., 1997*) | JAX ID: 004999 | |
| Strain, strain background (*Mus musculus, male*) | C57/BL/6-*Trdc*[tm1Mal]/J (Trcd-H2BeGFP) | Jackson Laboratory (*Prinz et al., 2006*) | JAX ID: 016941 | |
| Antibody | PerCP/C5.5 anti-mouse CD45 (rat monoclonal) | BioLegend | Cat.No.: 103131; RRID: AB_893344 | FC (1:200) for infection study |
| Antibody | AF700 anti-mouse CD3 (rat monoclonal) | BioLegend | Cat.No.: 100216; RRID: AB_493697 | FC (1:50) for infection study |
| Antibody | Brilliant Violet 511 anti-mouse/human CD45R/B220 (rat monoclonal) | BioLegend | Cat.No.: 103247; RRID: AB_2561394 | FC (1:200) for infection study |
| Antibody | Brilliant Violet 605 anti-mouse NK1.1 (mouse monoclonal) | BioLegend | Cat. No.: 108753; RRID: AB_2686977 | FC (1:200) for infection study |
| Antibody | Brilliant Violet 650 CD11c (Armenian Hamster monoclonal) | BioLegend | Cat.No.: 117339; RRID: AB_2562414 | FC (1:100) for infection study |
| Antibody | Brilliant Violet 711 anti-mouse Ly-6G/Ly-6C (GR-1) (rat monoclonal) | BioLegend | Cat.No.: 108443; RRID: AB_2562549 | FC (1:200) for infection study |
| Antibody | Brilliant Violet 785 anti-mouse I-A/I-E (MHC-II) (rat monoclonal) | BioLegend | Cat.No.: 107645; RRID: AB_2565977 | FC (1:200) for infection study |
| Antibody | PE/Dazzle594 anti-mouse F4/80 (rat monoclonal) | BioLegend | Cat.No.:123145; RRID: AB_2564132 | FC (1:100) for infection study |
| Antibody | PE/Cyanine7 anti-mouse CX3CR1 (mouse monoclonal) | BioLegend | Cat.No.:149015; RRID: AB_2565699 | FC (1:1000) for infection study |
| Antibody | FITC anti-mouse CCR2 (rat monoclonal) | BioLegend | Cat. No.: 150608; RRID: AB_2616980 | FC (1:200) for infection and steady-state study |
| Antibody | Brilliant Violet 421 anti-mouse CX3CR1 (mouse monoclonal) | BioLegend | Cat.No.: 149023; RRID: AB_2565706 | FC 1:1000 for steady-state study |
| Antibody | PerCP/Cyanine5.5 anti-mouse CD45 (rat monoclonal) | BioLegend | Cat.No.:157207; RRID: AB_2860727 | FC 1:100 for steady-state study |
| Antibody | PE/Cyanine7 anti-mouse F4/80 (rat monoclonal) | BioLegend | Cat.No.: 123113; RRID: AB_893490 | FC 1:100 for steady-state study |
| Antibody | APC anti-mouse CD163 (rat monoclonal) | BioLegend | Cat.No.:155305; RRID: AB_2814059 | FC 1:200 for steady-state study |
| Antibody | APC/Cy7 anti-mouse I-A/I-E (rat monoclonal) | BioLegend | Cat.No.: 107627; RRID: AB_1659252 | FC 1:200 for steady-state study |
| Antibody | Brilliant Violet 421 anti-mouse NK1.1 (mouse monoclonal) | BioLegend | Cat.No.: 108731; RRID: AB_10895916 | FC 1:200 for steady-state study |

*Appendix 1 Continued on next page*

*Appendix 1 Continued*

| Reagent type (species) or resource | Designation | Source or reference | Identifiers | Additional information |
|---|---|---|---|---|
| Antibody | FITC anti-mouse CD3 (rat monoclonal) | BioLegend | Cat.No.: 100203; RRID: AB_312660 | FC 1:50 for steady-state study |
| Antibody | PE anti-mouse B220 (CD45R) (rat monoclonal) | Miltenyi | Cat.No.: 130-120-077; RRID: AB_2751992 | 1:50 for steady-state study |
| Antibody | PE/Cyanine7 anti-mouse TCRbeta chain (Armenian Hamster monoclonal) | BioLegend | Cat.No. 109221; RRID: AB_893627 | 1:100 for steady-state study |
| Antibody | APC anti-mouse TCR g/d (Armenian Hamster monoclonal) | BioLegend | Cat.No.: 118115; RRID: AB_1731824 | 1:100 for steady-state study |
| Antibody | APC/Fire750 anti-mouse CD45 (mouse monoclonal) | BioLegend | Cat.No.: 103153; RRID: AB_2572115 | 1:100 for steady-state study |
| Antibody | Brilliant Violet 421 anti-mouse F4/80 (mouse monoclonal) | BioLegend | Cat.No.: 123131; RRID: AB_10901171 | 1:100 for steady-state study |
| Antibody | PE anti-mouse CD209a (DC-Sign) antibody (mouse monoclonal) | BioLegend | Cat.No.: 833003; RRID: AB_2721636 | FC 1:50 for steady-state study |
| Antibody | PerCP/Cy5.5 anti-mouse/human CD11b (rat monoclonal) | BioLegend | Cat.No.: 101228; RRID: AB_893232 | FC 1:200 for steady-state study |
| Antibody | APC anti-mouse I-A$^b$ (mouse monoclonal) | BioLegend | Cat.No:116418; RRID: AB_10574160 | FC 1:200 for steady-state study and parabiosis |
| Antibody | PerCP/Cyanine5.5 anti-mouse CD45.1 (mouse monoclonal) | BioLegend | Cat.No.: 110728; RRID: AB_893346 | FC 1:100 for steady-state study and parabiosis |
| Antibody | APC/Cyanine7 anti-mouse CD45.2 (mouse monoclonal) | BioLegend | Cat.No.: 109824; RRID: AB_830789 | i.v. injection 1:100 |
| Antibody | PE anti-mouse CD64 (FcγRI) (mouse monoclonal) | BioLegend | Cat.No:. 139303; RRID: AB_10612740 | FC 1:100 for steady-state study and parabiosis |
| Antibody | PE/Cyanine7 anti-mouse Tim-4 (rat monoclonal) | BioLegend | Cat.No.: 130010; RRID: AB_2565719 | FC 1:100 for parabiosis |
| Antibody | Brilliant Violet 785 anti-mouse/human CD11b (rat monoclonal) | BioLegend | Cat.No.: 101224; RRID: AB_755986 | FC 1:100 for parabiosis |
| Antibody | Alexa Fluor 700 anti-mouse Ly-6G (rat monoclonal) | BioLegend | Cat.No.: 127622; RRID: AB_10643269 | FC 1:100 for parabiosis |
| Antibody | PE anti-mouse CD115 (CSF-1R) (rat monoclonal) | BioLegend | Cat.No.: 135506; RRID: AB_1937253 | FC 1:100 for parabiosis |
| Antibody | FITC anti-mouse Ly-6C (rat monoclonal) | BioLegend | Cat.No.: 128006; RRID: AB_1186135 | FC 1:100 for parabiosis |
| Antibody | anti-mouse F4/80 (rat monoclonal) | BioLegend | Cat.No. MCA497G; RRID: AB_872005 | IF 1:200 |
| Antibody | anti-mouse Ly-6G+Ly-6C (rat monoclonal) | abcam | Cat.No. ab25377; RRID: AB_470492 | IF 1:500 |
| Antibody | Purified anti-mouse I-A/I-E (mouse monoclonal) | BioLegend | Cat.No.: 107601; RRID: AB_313316 | IF 1:200 |
| Antibody | Purified anti-mouse Ly-6C (rat monoclonal) | BioLegend | Cat.No. 128002 RRID: AB_1134214 | IF 1:100 |
| Antibody | Purified anti-mouse CD3 (mouse monoclonal) | BioLegend | Cat.No. 100202 RRID: AB_312659 | IF 1:50 |
| Antibody | Anti-mouse Clec9a (sheep polyclonal) | R&D Systems | Cat.No. AF6776 RRID: AB_10890771 | IF 1:50 |
| Antibody | anti-mouse CD163 [TNKUPJ] (rat monoclonal) | Invitrogen/ eBioscience | Cat.No. 14-1631-82 RRID: AB_2716934 | IF 1:200 |
| Antibody | anti-NCR1 antibody [EPR23097-35] (rabbit monoclonal) | abcam | Cat.No. ab233558 RRID: AB_2904203 | IF 1:50 |
| Antibody | Ani-mouse DC-Sign (DC28) (mouse monoclonal) | Santa Cruz | Cat.No. sc-65740 RRID: AB_1121347 | IF 1:50 |

*Appendix 1 Continued on next page*

*Appendix 1 Continued*

| Reagent type (species) or resource | Designation | Source or reference | Identifiers | Additional information |
|---|---|---|---|---|
| Antibody | Goat anti-rabbit IgG (H+L) Cross-Adsorbed Secondary Antibody, Alexa Fluor 488 | Invitrogen | Cat-No. 11008 RRID: AB_143165 | IF 1:2000 |
| Antibody | Goat anti-rabbit IgG (H+L) Cross-Adsorbed Secondary Antibody, Alexa Fluor 546 | Invitrogen | Cat.No. A-11010 RRID: AB_2534077 | IF 1:2000 |
| Antibody | Goat anti-rat IgG (H+L) Cross-Adsorbed secondary Antibody, Alexa Fluor 546 | Invitrogen | Cat.No. A-11081 RRID: AB_141738 | IF 1:2000 |
| Antibody | Goat Anti-Rat IgG H+L Alexa Fluor 647 | abcam | Cat.No. ab150159; RRID: AB_2566823 | IF 1:2000 |
| Antibody | Donkey Anti-Sheep IgG (H+L) Alexa fluor 546 | Invitrogen | Cat.No.:A-21098 RRID: AB_2535752 | IF: 1:2000 |
| Antibody | Brilliant Violet 421 Mouse IgG2a, κ Isotype Ctrl antibody | BioLegend | Cat.No.: 400259; RRID: AB_10895919 | FC 1:200 Isotype control |
| Antibody | Brilliant Violet 421 Rat IgG2a, κ Isotype Ctrl antibody | BioLegend | Cat.No.: 400535; RRID: AB_10933427 | FC 1:200 Isotype control |
| Antibody | Brilliant Violet 510 Rat IgG2a, κ, Isotype Ctrl antibody | BD Bioscience | Cat.No. 562952; RRID: AB_2869438 | FC 1:200 Isotype control |
| Antibody | FITC Rat IgG2b, κ Isotype Ctrl antibody | BioLegend | Cat.No.: 400605; RRID: AB_326549 | FC 1:200 Isotype control |
| Antibody | Alexa Fluor 488 Rat IgG2a, κ Isotype Ctrl antibody | BioLegend | Cat.No.: 400525; RRID: AB_2864283 | FC 1:200 Isotype control |
| Antibody | Alexa Fluor 488 Rat IgG2b, κ Isotype Ctrl antibody | BioLegend | Cat.No.: 400625; RRID: AB_389321 | FC 1:200 Isotype control |
| Antibody | PE Isotype Control Antibody, Rat IgG2a | Miltenyi | Cat.No.: 130-123-747; RRID: AB_2857628 | FC 1:200 Isotype control |
| Antibody | PE Rat IgG2a, κ Isotype Ctrl antibody | BioLegend | Cat.No.: 400507; RRID: AB_326530 | FC 1:200 Isotype control |
| Antibody | PE Mouse IgG2a, κ Isotype Ctrl antibody | BioLegend | Cat.No.: 400213; RRID: AB_2800438 | FC 1:200 Isotype control |
| Antibody | PE Rat IgG2b kappa Isotype Control | eBioscience | Cat.No.: 12-4031-82; RRID: AB_470042 | FC 1:200 Isotype control |
| Antibody | PerCP/Cyanine5.5, Rat IgG2b, κ Isotype Ctrl antibody | BioLegend | Cat.No. 400631; RRID: AB_893693 | FC 1:200 Isotype control |
| Antibody | PE/Cyanine 7 Mouse IgG1, κ Isotype Ctrl antibody | BioLegend | Cat.No.: 400125; RRID: AB_2861533 | FC 1:200 Isotype control |
| Antibody | PE/Cyanine 7 Rat IgG2a, κ Isotype Ctrl antibody | BioLegend | Cat.No.: 400521; RRID: AB_326542 | FC 1:200 Isotype control |
| Antibody | PE/Cyanine 7 Armenian Hamster IgG Isotype Ctrl antibody | BioLegend | Cat.No.:400921; RRID: AB_2905473 | FC 1:200 Isotype control |
| Antibody | APC Rat IgG2a, κ Isotype Ctrl antibody | BioLegend | Cat.No.: 400511; RRID: AB_2814702 | FC 1:200 Isotype control |
| Antibody | APC Armenian Hamster IgG Isotype Ctrl antibody | BioLegend | Cat.No.: 400911; RRID: AB_2905474 | FC 1:200 Isotype control |
| Antibody | APC Mouse IgG2a, κ Isotype Ctrl (FC) antibody | BioLegend | Cat.No.: 400221; RRID: AB_2891178 | FC 1:200 Isotype control |
| Antibody | APC/Cyanine7 Rat IgG2b, κ Isotype Ctrl antibody | BioLegend | Cat.No.: 400628; RRID: AB_326565 | FC 1:200 Isotype control |
| Antibody | APC/Fire750 Rat IgG2b, κ Isotype Ctrl antibody | BioLegend | Cat.No.: 400669; RRID: AB_2905475 | FC 1:200 Isotype control |
| Commercial assay or kit | M.O.M. (Mouse on Mouse) Immunodetection Kit | Vector Laboratories | Cat.No. BMK-2202 | |
| Commercial assay or kit | RNeasy Mini Kit | Qiagen | Cat.No.: 74004 | |

*Appendix 1 Continued on next page*

*Appendix 1 Continued*

| Reagent type (species) or resource | Designation | Source or reference | Identifiers | Additional information |
|---|---|---|---|---|
| Commercial assay or kit | LEGENDPlex with Mouse Inflammation Panel | BioLegend | Cat.No.: 740446 | |
| Commercial assay or kit | SMARTer Stranded Total RNA-Seq Kit – Pico Input Mammalian | Takara | Cat.No.: 634488 | |
| Chemical compound, drug | Collagenase D from *Clostridium histolyticum* | Roche | Cat.No. 11088858001 | |
| Chemical compound, drug | DNase I | Sigma | Cat.No. D4513 | |
| Chemical compound, drug | Hyaluronidase type I-S | Sigma | Cat.No. H3506 | |
| Chemical compound, drug | RBC Lysis Solution | Qiagen | Cat.No.: 158904 | |
| Chemical compound, drug | UltraPure Lipopolysaccharide from *Escherichia coli* O55:B5 | Sigma | Cat.No.: L2880 | |
| Chemical compound, drug | Gibco RPMI1640 media | Fisher Scientific | Cat-No.: 11530586 | |
| Chemical compound, drug | QIAzol Lysis Reagent | Qiagen | Cat.No.: 79306 | |
| Chemical compound, drug | Fc blocking reagent | Miltenyi | Cat.No. 130-092-575 | |
| Chemical compound, drug | Gentamicin solution | Sigma/ Merck | Cat.No. G1397 | |
| Other (dyes) | ZombieAqua | BioLegend | Cat.No.: 423101 | |
| Other (dyes) | ZombieNIR | BioLegend | Cat. No.: 423105 | |
| Other (dyes) | Viobility 405/452 Fixable Dye | Miltenyi | Cat.No.: 130-092-575 | |
| Other (dyes) | DAPI | Invitrogen | D1306 | 1 µg/ml |
| Other | ProLong Antifade Gold with DAPI | Invitrogen | P36931 | |
| Other | ProLong Antifade Gold w/o DAPI | Invitrogen | P36930 | |
| Software, algorithm | FlowJo v10.8.2 | BD Life Sciences | RRID: SCR_008520 | https://www.flowjo.com/ |
| Software, algorithm | Adobe Illustrator 2020 (v24.0.1) | Adobe | RRID: SCR_010279 | https://www.adobe.com/de/products/illustrator.html |
| Software, algorithm | GraphPad Prism v5 | GraphPad Software | RRID: SCR_002798 | https://www.graphpad.com/scientific-software/prism/ |
| Software, algorithm | InkScape v0.92.4 | The Inkscape Project | RRID: SCR_014479 | https://inkscape.org/de/release/inkscape-0.92.4/ |
| Software, algorithm | ImageJ 1.53 a | Wayne Rasband National Institute of Health, USA | RRID: SCR_003070 | https://imagej.nih.gov/ij/index.html |
| Software, algorithm | Zen 2.3 Version 14.0.26.201 | Carl Zeiss Microscopy | | https://www.zeiss.de/mikroskopie/produkte/mikroskopsoftware/zen-lite/zen-lite-download.html |
| Software, algorithm | LEGENDPlex Software v8.0 | BioLegend | software provied by BioLegend as part of the LegendPlex kit for protein analysis | https://www.biolegend.com/en-us/legendplex |
| Software, algorithm | FastQC | *Andrews, 2010* | RRID: SCR_014583 | http://www.bioinformatics.babraham.ac.uk/projects/fastqc/ |
| Software, algorithm | STAR 2.6.1d | *Dobin et al., 2013* | RRID: SCR_004463 | https://github.com/alexdobin/STAR/releases?page=2 |

*Appendix 1 Continued on next page*

*Appendix 1 Continued*

| Reagent type (species) or resource | Designation | Source or reference | Identifiers | Additional information |
|---|---|---|---|---|
| Software, algorithm | Subread package | *Liao et al., 2013* | RRID: SCR_009803 | http://subread.sourceforge.net/ |
| Software, algorithm | DESeq2 V1.18.1 | *Love et al., 2014* | RRID: SCR_015687 | https://bioconductor.org/packages/release/bioc/html/DESeq2.html |
| Other | BD Aria Fusion | BD BioScience | N/A | Instrument |
| Other | Homogenizer MM400 | Retsch | N/A | Instrument |
| Other | LabChip Gx Touch 24 | Perkin Elmer | N/A | Instrument |
| Other | Leica Cryotome CM1850 | Leica | N/A | Instrument |
| Other | Leica Microtome RM2255 | Leica | N/A | Instrument |
| Other | MACS Quant Analyzer 10 | Miltenyi | N/A | Instrument |
| Other | BD LSRFortessa Cell Analyzer | BD Biosciences | N/A | Instrument |
| Other | NextSeq500 | Illumina | N/A | Instrument |
| Other | Zeiss LSM710 Confocal Microscope | Carl Zeiss Microscopy | N/A | Instrument |
| Other | Olympus BX51 | Olympus | N/A | instrument |

