## [Editor Report]

This manuscript reports important findings regarding the highly variable immune environments along the epididymis. Using multiple mouse models (bacterial infection and parabiosis between WT and Ccr2 KO) in conjunction with scRNA-seq analyses, the authors provided solid evidence supporting the notion that resident immune cells are strategically positioned along the epididymal duct, potentially providing different immunological environments required for sperm maturations and elimination of pathogens ascending the urogenital tract.

---

## [Decision Letter]

**Decision letter after peer review:**

Thank you for submitting your article "The regional distribution of resident immune cells shapes distinct immunological environments along the murine epididymis" for consideration by *eLife*. Your article has been reviewed by 2 peer reviewers, one of whom is a member of our Board of Reviewing Editors, and the evaluation has been overseen by Ricardo Azziz as the Senior Editor. The reviewers have opted to remain anonymous.

Essential revisions:

1) Fix the problems identified in the flow cytometry and immunofluorescent data.

2) Tone down the claim and revise the Abstract.

*Reviewer #1 (Recommendations for the authors):*

Please address the following concerns:

1) The strategic positioning of resident immune cells should function in two aspects: facilitating sperm maturation and infection prevention. The data were focused on the latter. Any evidence for the first function?

2) In the bacterial infection model, UPEC was injected from the vas deferens. Why not inject from the caput side, e.g., through efferent ducts or rete testis? it would be ideal to do this instead of in vitro culture as it is more physiological.

3) Abstract does not reflect what had been done in this study and thus, should be revised to include all major experiments and their collective conclusion.

4) Does this study have any clinical implications? Ideally, this should be briefly discussed in the Discussion section.

*Reviewer #2 (Recommendations for the authors):*

Abstract: Please provide more information about the data presented in this study. As it is currently written the abstract only refers to the single-cell RNA sequencing data in the steady state epididymis.

1) Flow cytometry experiments) In Figure 2D, the authors should show quantification of the percentages of immune cell subsets relative to live cells, similar to the data shown in Figure 2B for CD45+ cells. As it is, this figure does not support the statement that "Infiltration of neutrophils was followed by an influx of monocytes (Ly6C+ CD11b+, Figure 2D)". In order to conclude that there was infiltration of these immune cells, the authors should provide quantification of neutrophils and monocytes relative to the total number of live cells analysed. In addition, both these cell populations appear to rise concomitantly following bacterial injection, which does not support the conclusion that monocyte recruitment follows neutrophil recruitment.

2) Figure 4A: flow cytometry analysis) The data showing higher numbers of CD45+ cells relative to live cells in the IS versus the more distal epididymal segments should be discussed with respect to previous studies. Please specify the region shown in the right panel's IS/CT and CS/CD regions. This is especially important for the IS and CT, which have distinct morphological appearances.

3) Figure 6C and D: A surprisingly low number of CX3CR1-EGFP cells was detected by immunofluorescence in the cauda. This is not in agreement with previous studies showing a similar % of CX3CR1-EGFP cells in the IS and cauda regions by immunofluorescence and flow cytometry. The authors need to discuss this discrepancy. Perhaps the different fixation procedures used in the current study compared to those used in previous studies could account for the loss of EGFP in the epididymis sections. As such, cells that appear to be F4/80 positive but negative for EGFP by immunofluorescence might simply be due to the loss of cytoplasmic EGFP, while F4/80 immunogenicity remained intact (line 415).

4) Line 265: The strategy to exclude vascular CD45+ cells should be mentioned in the Results section.

5) Line 315: Please provide a reference to support this statement "Adgre1+C1qa+ cells, broadly considered as epididymal macrophages, constitute the majority of CD45+ cells in the epididymis".

6) Line 328 and Figure 5C: The authors state: Adgre1+C1qa+ cells were re-analyzed after exclusion of other CD45+ cells". They then state: "All identified macrophage subgroups were highly enriched with C1qa and Adgre1 transcripts confirming their macrophage identity (Figure 5D, Figure S5A)." I may have missed something but is it not an obvious result since they had initially selected cells using these markers?

7) In the Discussion section (line 475), the following statement: "(…) we further demonstrate that the accompanying leukocytic infiltration is characterized by a massive influx of neutrophils and monocyte-derived MHC-IIhi macrophages" is not supported by the data. As stated above, to make this conclusion, flow cytometry analysis of the percentage of neutrophils and macrophages relative to the total number of live cells should be provided.

8) Line 528: Please cite Mendelsohn et al. AJP Cell Physiol. 2020.

9) Line 531: The authors state: "Intriguingly, our data revealed that distinct immunological landscapes exist within proximal (IS, caput) and distal regions (corpus, cauda), that are tailored to the respective needs of the microenvironments." However, they should acknowledge that their results only reinforce previous studies that have already shown the presence of immune cells with markedly distinct phenotypes in the different regions of the epididymis. The way this sentence is written at the moment, the authors seem to imply that this is the first study that describes immune cell heterogeneity in this organ.

10) Line 534: The conclusion that macrophages constitute the major immune cell population of the murine epididymis is not supported by the data provided here. In fact, the authors found that macrophages account for only approximately 20% of CD45+ immune cells in the cauda (Figure 4B). The authors should, therefore, modify their conclusion to state that macrophages constitute the major immune cell population in the IS. In fact, this conclusion would be more in line with previously published studies.

11) Line 553: Here again, the statement that the transcriptional profile of CX3CR1+ cells indicates a macrophage phenotype is an over-simplification. Please mention the study by Battistone et al. (MHR 2020) that characterized CX3CR1-positive cells in all segments of the epididymis. In the Battistone study, while several CX3CR1-EGFP+ cells were described as having a macrophage phenotype, some cells had a dendritic cell phenotype, especially in the cauda. The current study actually confirms the previously published higher number of cells with a macrophage phenotype in the IS versus other regions.

12) Line 555: Please be careful with the statement that fewer intraepithelial CX3CR1-EGFP+ cells are present in the cauda. How were these intraepithelial cells quantified? Here again, the fact that fewer EGFP+ cells were observed in the cauda versus other regions is not in agreement with previous studies. The authors should discuss their results with respect to previous studies indicating that a similar percentage of CX3CR1-EGFP+ cells (with respect to the number of live cells analysed) were detected in the IS and cauda regions, while a higher percentage was detected in the cauda versus the caput/corpus regions (Battistone MHR 2020).

Other specific comments:

Line 72-76: the proximal regions appear to be almost unresponsive to which stimuli?

Lines 136-137: Please mention that a low level of histopathological damage in the caput 10 days post-UPEC injection has been reported previously (Klein et al. MHR 2020).

Figure 1E: Would it be possible to label bacteria for microscopic assessment in these tissues? If so, the authors should provide these data.

Figure 2A: Please describe how immune cells were identified and how the area of immune cell infiltrates was quantified.

Figure 4B-H: Please provide a list of the markers that were used to identify the different cell types by flow cytometry.

Figure 6C and line 388: Please change "principal cell" by "epithelial cells" since no marker was used to identify these cell types. Previous studies have indicated that immune cells can send their projections not only between principal cells but also next to narrow cells in the IS.

Method Gating of immune cells under physiological conditions: It is stated that CX3CR1 was used as a positive marker of macrophages. However, previous studies showed that dendritic cells can also express this marker. This caveat should be mentioned.

---

## [Author Response]

Reviewer #1 (Recommendations for the authors):Please address the following concerns:1) The strategic positioning of resident immune cells should function in two aspects: facilitating sperm maturation and infection prevention. The data were focused on the latter. Any evidence for the first function?

In the present study, we were aiming at assessing the murine epididymal immune cell diversity and their transcriptional identity at full width. In this regard, we were able to show that distinct populations reside within the epididymis and reveal striking differences in their regional distribution. At this current stage, we cannot provide further experimental evidence for the direct or indirect influence of particular immune cell populations on sperm maturation. As stated in the public comment, this would require several comprehensive experimental approaches using transgenic mouse models in which particular immune cell populations are selectively depleted followed by functional sperm analysis (i.e. motility/ vitality assessment, capacitation and fertilization competencies) and assessment of putative inflammatory responses (formation of anti-sperm antibodies, region-specific expression of inflammatory mediators).

Yet, our data strongly support that resident immune cells differently populate the epididymal regions and are likely involved in tissue maintenance. Based on the transcriptional identity of the identified cell types combined with the current knowledge of function of particular immune cell subpopulations derived from other organs combined with their location within the epididymal regions (intraepithelial/ periductal/ interstitial), we are confident to predict their general function (homeostatic vs. inflammatory). For example, combining our data with previous reports, especially on intraepithelial CX3CR1^+^ cells (Smith et al., 2014; Battistone et al., 2020), evidence is given that these cells are indispensable for maintaining epithelial integrity, that in turn, is mandatory for preserving the luminal microenvironment and potentially protecting luminal spermatozoa from autoimmune reactions.

To address the reviewer’s point, we have extended the respective part about intraepithelial CX3CR1^+^ cells in the discussion (line 573-575) stating that the maintenance of epithelial integrity is mandatory to maintain the luminal microenvironment required for proper sperm maturation. In line 579-580, we have included a sentence stating that it remains elusive whether these immune cells also have a direct impact on sperm maturation.

2) In the bacterial infection model, UPEC was injected from the vas deferens. Why not inject from the caput side, e.g., through efferent ducts or rete testis? it would be ideal to do this instead of in vitro culture as it is more physiological.

Our main experimental model is based on the bilateral intravasal injection of uropathogenic *Escherichia coli* (UPEC). This administration route mimics the clinical situation best as in bacterial epididymitis (in contrast to viral orchitis) urogenital pathogens are ascending canalicularly through the male urogenital tract.

In the pathological setting, bacteria do not settle the testis first and then ‘descend’ from the testis to the epididymis. Moreover, it would technically be very demanding to first expose, then inject into the efferent ducts or rete testis both of which have a considerably smaller lumen than the vas deferens and particularly in the latter case an interstitial injection cannot be excluded.

The aim of the *ex vivo* model was to confirm that the observed regional differences in our approach is not only due to the ascending nature of the model and thus, a longer exposure of the cauda to the pathogens compared to other regions. An injection into the proximal sites, e.g. efferent duct or rete testis (instead of injecting into the distal sites), would therefore not answer this particular question as another gradient would be generated (high pathogen concentration in the proximal part and low concentration in the distal part). The advantage of the ex vivo model is the simultaneous exposure to the inflammatory insult which excludes variable time or magnitude of stimulus.

Of note, a region-specific magnitude of immune response independent of the administration route was also seen in models by other groups by employing systemic inflammation (Wang et al., 2019) as well as an autoimmune-based disease model (Wijayarathna et al., 2020).

3) Abstract does not reflect what had been done in this study and thus, should be revised to include all major experiments and their collective conclusion.

We have rephrased the abstract and included more details about the experimental approaches.

4) Does this study have any clinical implications? Ideally, this should be briefly discussed in the Discussion section.

As the current manuscript is designed as a tool and resource paper that serves as the basis for further experimental approaches to assess function of particular immune cell populations, the study does not have concrete clinical implication. However, another publication of our group (Klein et al., 2019) has demonstrated that a combined treatment with antibiotics and dexamethasone dampens the magnitude of the immune response and thus, reduces tissue damage within the cauda.

Reviewer #2 (Recommendations for the authors):Abstract: Please provide more information about the data presented in this study. As it is currently written the abstract only refers to the single-cell RNA sequencing data in the steady state epididymis.

We have rephrased the abstract and included more details about the experimental approaches.

1) Flow cytometry experiments) In Figure 2D, the authors should show quantification of the percentages of immune cell subsets relative to live cells, similar to the data shown in Figure 2B for CD45+ cells. As it is, this figure does not support the statement that "Infiltration of neutrophils was followed by an influx of monocytes (Ly6C+ CD11b+, Figure 2D)". In order to conclude that there was infiltration of these immune cells, the authors should provide quantification of neutrophils and monocytes relative to the total number of live cells analysed. In addition, both these cell populations appear to rise concomitantly following bacterial injection, which does not support the conclusion that monocyte recruitment follows neutrophil recruitment.

We have replaced the data shown in Figure 2 by a new high-dimensional flow cytometry analysis including FtlSNE visualization of the CD45^+^ cell population in epididymal regions (IS, Caput, Corpus, Cauda) in naive, sham and UPEC mice, as well as bar diagrams showing the percentage in single live cells of the respective populations. In contrast to the previous version, the current data do not show the disease time course, but focus on day 10 after infection (similarly to the RNASeq data in Figure 1).

2) Figure 4A: flow cytometry analysis) The data showing higher numbers of CD45+ cells relative to live cells in the IS versus the more distal epididymal segments should be discussed with respect to previous studies.

Please see the response to the respective public comment (comment 3 in the public review) where we have addressed this point.

Please specify the region shown in the right panel's IS/CT and CS/CD regions. This is especially important for the IS and CT, which have distinct morphological appearances.

We have changed the labeling of Figure 4A accordingly. The images are displaying CD45^+^ cells within the IS and within the corpus.

3) Figure 6C and D: A surprisingly low number of CX3CR1-EGFP cells was detected by immunofluorescence in the cauda. This is not in agreement with previous studies showing a similar % of CX3CR1-EGFP cells in the IS and cauda regions by immunofluorescence and flow cytometry. The authors need to discuss this discrepancy. Perhaps the different fixation procedures used in the current study compared to those used in previous studies could account for the loss of EGFP in the epididymis sections. As such, cells that appear to be F4/80 positive but negative for EGFP by immunofluorescence might simply be due to the loss of cytoplasmic EGFP, while F4/80 immunogenicity remained intact (line 415).

Please see the response to the respective public comment (comment 3 in the public review), where we have addressed this point.

4) Line 265: The strategy to exclude vascular CD45+ cells should be mentioned in the Results section.

We have included a brief description of the strategy connecting to the respective figures (Figure 3A, figure 3 – Supplement 1). Hopefully, this will facilitate understanding of the strategy so that the reader may not need to refer to the full description in the methods section.

5) Line 315: Please provide a reference to support this statement "Adgre1+C1qa+ cells, broadly considered as epididymal macrophages, constitute the majority of CD45+ cells in the epididymis".

We are unable to provide a reference for the statement that Adgre1^+^C1qa^+^ cells are epididymal macrophages as this is the first study that unravels the full transcriptional profile and identity of closely related subpopulations of the mononuclear phagocyte system within the murine epididymis. Our statement is based on the fact that both Adgre and C1qa are well-known and accepted key macrophage markers. We have included the reference (Dick et al., 2022). This reference reports on the macrophage profile incl. key markers conserved among organs.

6) Line 328 and Figure 5C: The authors state: Adgre1+C1qa+ cells were re-analyzed after exclusion of other CD45+ cells". They then state: "All identified macrophage subgroups were highly enriched with C1qa and Adgre1 transcripts confirming their macrophage identity (Figure 5D, Figure S5A)." I may have missed something but is it not an obvious result since they had initially selected cells using these markers?

Thank you for this comment. For this approach, we have not performed a marker-based selection, but we have re-analyzed cluster 1, 2, 7 under exclusion of all other clusters.

For better clarity, we have reworded the respective sentence to avoid misunderstanding. See line: 352-353:

“… all cells in clusters 1, 2 and 7 were re-analyzed after exclusion of other CD45^+^ cells.

7) In the Discussion section (line 475), the following statement: "(…) we further demonstrate that the accompanying leukocytic infiltration is characterized by a massive influx of neutrophils and monocyte-derived MHC-IIhi macrophages" is not supported by the data. As stated above, to make this conclusion, flow cytometry analysis of the percentage of neutrophils and macrophages relative to the total number of live cells should be provided.

The critic is valid. As stated above, we have replaced data shown previously in Figure 2 by a high-dimensional flow cytometry approach with FltSNE visualization and bar diagrams now displaying the population in relation to single live cells (revised Figure 2)

8) Line 528: Please cite Mendelsohn et al. AJP Cell Physiol. 2020.

We have included Mendelsohn et al. AJP Cell Physiol, 2020 to the listed references.

9) Line 531: The authors state: "Intriguingly, our data revealed that distinct immunological landscapes exist within proximal (IS, caput) and distal regions (corpus, cauda), that are tailored to the respective needs of the microenvironments." However, they should acknowledge that their results only reinforce previous studies that have already shown the presence of immune cells with markedly distinct phenotypes in the different regions of the epididymis. The way this sentence is written at the moment, the authors seem to imply that this is the first study that describes immune cell heterogeneity in this organ.

In order to accommodate the critique, we have rephrased the statement as follows:

“our data unraveled the transcriptional identity and tissue location of extravascular immune cells and further support the existence of distinct immunological environments along the epididymal duct that are tailored to the respective needs of the microenvironment” within the Discussion section (line 555-558)”.

Please see also our comment to a similar point in the public review.

10) Line 534: The conclusion that macrophages constitute the major immune cell population of the murine epididymis is not supported by the data provided here. In fact, the authors found that macrophages account for only approximately 20% of CD45+ immune cells in the cauda (Figure 4B). The authors should, therefore, modify their conclusion to state that macrophages constitute the major immune cell population in the IS. In fact, this conclusion would be more in line with previously published studies.

Please see the response in the respective comment within the public review:

We fully agree with the reviewer and have changed the conclusion to “macrophages constitute the major immune cell population, especially in the IS” (line 559-560).

11) Line 553: Here again, the statement that the transcriptional profile of CX3CR1+ cells indicates a macrophage phenotype is an over-simplification. Please mention the study by Battistone et al. (MHR 2020) that characterized CX3CR1-positive cells in all segments of the epididymis. In the Battistone study, while several CX3CR1-EGFP+ cells were described as having a macrophage phenotype, some cells had a dendritic cell phenotype, especially in the cauda. The current study actually confirms the previously published higher number of cells with a macrophage phenotype in the IS versus other regions.

A classification of these cells based on their structure presents not a very suitable tool to discriminate it from dendritic cells. This is exemplified by macrophages of the testis where two distinct subpopulations of macrophages exist with distinct morphologies (peritubular macrophages: flat and stellate shape similar to DC interstitial macrophages: more compact with partial exhibition of protrusions). Another example are CX3CR1-expressing microglia, highly specialized macrophages of the brain that exhibit a very typical stellate morphology (again similar to DC) in shape that is required for their homeostatic and sensing function. Therefore, the shape of a mononuclear phagocyte is not a reliable criteria for assessing their identity. For this purpose, we decided to perform a scRNASeq approach as a most powerful tool to identify a differential transcriptional profile of extravascular CD45^+^ cells, thus allowing a discrimination into classes and subtypes of mononuclear cells. Following clustering and analysis of the gene expression profiles, we have categorized these cells as macrophages characterized by their transcriptional profile, i.e. expression of established macrophage markers [C1qa, Adgre1, Fcgr1] with concomitant lack of DC markers [Flt3, Clec9a Cd209a]. Consequently, we regard these cells as clearly distinct from the identified dendritic cell populations (data shown in Figure 3)

As stated in the manuscript, intraepithelial CX3CR1^+^ macrophages of the epididymis actually possess a microglia-like transcriptional profile (see figure 5E and respective text passage). Therefore, it is plausible that these cells exhibit protrusions that facilitate a sampling function.

12) Line 555: Please be careful with the statement that fewer intraepithelial CX3CR1-EGFP+ cells are present in the cauda. How were these intraepithelial cells quantified? Here again, the fact that fewer EGFP+ cells were observed in the cauda versus other regions is not in agreement with previous studies. The authors should discuss their results with respect to previous studies indicating that a similar percentage of CX3CR1-EGFP+ cells (with respect to the number of live cells analysed) were detected in the IS and cauda regions, while a higher percentage was detected in the cauda versus the caput/corpus regions (Battistone MHR 2020).

Please see the response to the respective comment within the public review.

Other specific comments:Line 72-76: the proximal regions appear to be almost unresponsive to which stimuli?

In the prior sentence, it is stated:

“Previous investigations in rodents revealed differences in the immune reactions at the opposing ends of the epididymis following ascending bacterial infection and other inflammatory stimuli. In this regard, the proximal regions appear to be almost unresponsive …”.

We intended to interpret the mentioned sentence in context with the previous sentence.

For the sake of better clarity, we have now included the phrase “local and systemic inflammatory stimuli” in line 87 to point out the fundamental differences independent of the administration route of the stimulation.

Lines 136-137: Please mention that a low level of histopathological damage in the caput 10 days post-UPEC injection has been reported previously (Klein et al. MHR 2020).

We have included the phrase “in line with previous reports (Klein et al., 2020), ….” (see line 149-150).

Figure 1E: Would it be possible to label bacteria for microscopic assessment in these tissues? If so, the authors should provide these data.

The diagrams in the supplemental figures show bacterial numbers in association with neutrophil numbers as a surrogate as we were unsuccessful in our efforts to immunostain for *E. coli*.

While these diagrams were included in Figure S1 of the previous version, these can now be found in Figure 2 –Supplement 2.

Figure 2A: Please describe how immune cells were identified and how the area of immune cell infiltrates was quantified.

For morphometric assessment, we have stained the epididymal tissue by Masson-Goldner staining that allows a discrimination of distinct cell types (due to the trichrome system). Using this staining, immune cell infiltrates can be recognised distinctly (Visible in the representative histology images) and were measured using imageJ (see line 1076-1079). We are aware that this is only a semi-quantitative approach, but find it useful to link the flow cytometry analysis (that does not allow a localization of all immune cells) with histopathological observations.

Figure 4B-H: Please provide a list of the markers that were used to identify the different cell types by flow cytometry.

Previously, this information was scattered in the legends of the subpanels (e.g. total macrophages [F4/80^+^] …). We have now included a list of markers into the figure legend 4B-H onwards and agree that this list is useful in providing the reader with all required information without going to other sections.

Figure 6C and line 388: Please change "principal cell" by "epithelial cells" since no marker was used to identify these cell types. Previous studies have indicated that immune cells can send their projections not only between principal cells but also next to narrow cells in the IS.

We fully agree with the reviewer and have changed the wording from “adjacent principal cells” to “adjacent epithelial cells” in line 411-412.

Method Gating of immune cells under physiological conditions: It is stated that CX3CR1 was used as a positive marker of macrophages. However, previous studies showed that dendritic cells can also express this marker. This caveat should be mentioned.

Generally, CX3CR1 is a receptor that can be expressed by several myeloid cell populations. Initial observations indeed suggested that CX3CR1 is also expressed on the surface of dendritic cells and is required for their function and development (Łyszkiewicz et al., 2011). However, more recent experiments have revealed that CX3CR1 is exclusively expressed by monocyte precursors and monocyte-derived DC (Sutti et al., 2015; Sutti et al., 2019) which represents a developmentally distinct DC population only present during inflammation (Bosteels et al., 2020).

In terms of epididymal mononuclear phagocytes, the identity of resident CX3CR1^+^ cells remained for a long time not fully clarified (as stated within the discussion of the present manuscript). CX3CR1^+^ cells (at least a fraction of this heterogeneous cell pool) in the epididymis were originally classified as ‘dendritic cells’ due to their shpae and later more generally as ‘mononuclear phagocytes’. A major outcome of our study is that our scRNASeq data strongly point out that these cells possess a macrophage identity, an aspect that is stated in the discussion and based on the transcriptional profile and expression of key macrophage lineage markers.

This conclusion based on data seen in Figure 3D showing the expression of Cx3cr1 in macrophage clusters [1, 2, 7], less in monocyte clusters [8, 10] and barely in DC clusters is also in line with the above mentioned publications ((Bosteels et al., 2020) Therefore, we have integrated CX3CR1 in a panel of markers that also included CD45, F4/80, MHC-II, CD11b, CCR2, CD163) for flow cytometry analysis of macrophages, avoiding the use of CX3CR1 as a single positive marker. This panel was designed based on our single cell RNA sequencing data in which several markers were identified to be critical for separating closely related subpopulations (see Figure 3 and 5). Further information on the gating are shown in the respective supplemental figure.

References

Battistone, M.A., Mendelsohn, A.C., Spallanzani, R.G., Brown, D., Nair, A.V., and Breton, S. (2020). Region-specific transcriptomic and functional signatures of mononuclear phagocytes in the epididymis. Molecular human reproduction 26, 14-29.

Bosteels, C., Neyt, K., Vanheerswynghels, M., van Helden, M.J., Sichien, D., Debeuf, N., Prijck, S. de, Bosteels, V., Vandamme, N., and Martens, L., et al. (2020). Inflammatory Type 2 cDCs Acquire Features of cDC1s and Macrophages to Orchestrate Immunity to Respiratory Virus Infection. Immunity 52, 1039-1056.e9.

Dick, S.A., Wong, A., Hamidzada, H., Nejat, S., Nechanitzky, R., Vohra, S., Mueller, B., Zaman, R., Kantores, C., and Aronoff, L., et al. (2022). Three tissue resident macrophage subsets coexist across organs with conserved origins and life cycles. Science immunology 7, eabf7777.

Klein, B., Pant, S., Bhushan, S., Kautz, J., Rudat, C., Kispert, A., Pilatz, A., Wijayarathna, R., Middendorff, R., and Loveland, K.L., et al. (2019). Dexamethasone improves therapeutic outcomes in a preclinical bacterial epididymitis mouse model. Human reproduction (Oxford, England) 34, 1195-1205.

Łyszkiewicz, M., Witzlau, K., Pommerencke, J., and Krueger, A. (2011). Chemokine receptor CX3CR1 promotes dendritic cell development under steady-state conditions. Eur. J. Immunol. 41, 1256-1265.

Smith, T.B., Cortez-Retamozo, V., Grigoryeva, L.S., Hill, E., Pittet, M.J., and Da Silva, N. (2014). Mononuclear phagocytes rapidly clear apoptotic epithelial cells in the proximal epididymis. Andrology 2, 755-762.

Sutti, S., Bruzzì, S., Heymann, F., Liepelt, A., Krenkel, O., Toscani, A., Ramavath, N.N., Cotella, D., Albano, E., and Tacke, F. (2019). CX3CR1 Mediates the Development of Monocyte-Derived Dendritic Cells during Hepatic Inflammation. Cells 8.

Sutti, S., Locatelli, I., Bruzzì, S., Jindal, A., Vacchiano, M., Bozzola, C., and Albano, E. (2015). CX3CR1-expressing inflammatory dendritic cells contribute to the progression of steatohepatitis. Clinical science (London, England : 1979) 129, 797-808.

Voisin, A., Whitfield, M., Damon-Soubeyrand, C., Goubely, C., Henry-Berger, J., Saez, F., Kocer, A., Drevet, J.R., and Guiton, R. (2018). Comprehensive overview of murine epididymal mononuclear phagocytes and lymphocytes: Unexpected populations arise. Journal of reproductive immunology 126, 11-17.

Wang, F., Liu, W., Jiang, Q., Gong, M., Chen, R., Wu, H., Han, R., Chen, Y., and Han, D. (2019). Lipopolysaccharide-induced testicular dysfunction and epididymitis in mice: a critical role of tumor necrosis factor α†. Biology of reproduction 100, 849-861.

Wijayarathna, R., Pasalic, A., Nicolas, N., Biniwale, S., Ravinthiran, R., Genovese, R., Muir, J.A., Loveland, K.L., Meinhardt, A., and Fijak, M., et al. (2020). Region-specific immune responses to autoimmune epididymitis in the murine reproductive tract. Cell and tissue research.